**Applying the Dark Target aerosol algorithm with Advanced Himawari Imager**
**observations during the KORUS-AQ field campaign**
Pawan Gupta[1, 2], Robert C. Levy[3], Shana Mattoo[3, 4], Lorraine A. Remer[5], Robert E. Holz[6],
Andrew K. Heidinger[7]
[1] {STI, Universities Space Research Association (USRA), Huntsville, AL, USA}
[2] {NASA Marshall Space Flight Center, Huntsville, AL 35758, USA}
[3] {NASA Goddard Space Flight Center, Greenbelt, MD 20771, USA}
[4] {Science Systems and Applications, Inc, Lanham, MD 20709, USA}
[5] {JCET, University of Maryland – Baltimore County, Baltimore, MD 21228, USA}
[6] {SSEC, University of Wisconsin-Madison, WI 53707, USA}
[7] {NOAA Advanced Satellite Product Branch, Madison, WI 53707, USA}
Correspondence to: Pawan Gupta (pawan.gupta@nasa.gov)
**Abstract**
For nearly two decades we have been quantitatively observing the Earth's aerosol system
from space at one or two times of the day by applying the Dark Target family of
algorithms to polar-orbiting satellite sensors, particularly MODIS and VIIRS. With the
launch of the Advanced Himawari Imager (AHI) and the Advanced Baseline Imagers
(ABIs) into geosynchronous orbits, we have the new ability to expand temporal coverage
of the traditional aerosol optical depth (AOD) to resolve the diurnal signature of aerosol
loading during daylight hours. The Korean-United States – Air Quality (KORUS-AQ)
campaign taking place in and around the Korean peninsula during May-June 2016
initiated a special processing of full disk AHI observations that allowed us to make a
preliminary adoption of Dark Target aerosol algorithms to the wavelengths and
resolutions of AHI. Here, we describe the adaptation and show retrieval results from AHI
for this two-month period. The AHI-retrieved AOD is collocated in time and space with
existing AErosol RObotic NETwork stations across Asia and with collocated Terra- and
Aqua-MODIS retrievals. The new AHI AOD product matches AERONET, as well as
does the standard MODIS product, and the agreement between AHI and MODIS
retrieved AOD is excellent, as can be expected by maintaining consistency in algorithm
architecture and most algorithm assumptions. Furthermore, we show that the new product
approximates the AERONET-observed diurnal signature. Examining the diurnal patterns
of the new AHI AOD product we find specific areas over land where the diurnal signal is
spatially cohesive. For example, in Bangladesh the AOD increases by 0.50 from morning
to evening, and in northeast China the AOD decreases by 0.25. However, over open
ocean the observed diurnal cycle is driven by two artifacts, one associated with solar
zenith angles greater than $70^o$ that may be caused by a radiative transfer model that does
not properly represent spherical Earth, and the other artifact associated with the fringes of
the $40^o$ glint angle mask. This opportunity during KORUS-AQ provides encouragement
to move towards an operational Dark Target algorithm for AHI. Future work will need to
re-examine masking including snow mask, re-evaluate assumed aerosol models for
geosynchronous geometry, address the artifacts over the ocean and investigate size
parameter retrieval from the over ocean algorithm.
**1.0 Introduction**
Atmospheric aerosols, small liquid or solid particles suspended in the atmosphere, play a
key role in Earth's energy balance, cloud physics, geochemical cycles and air
quality/public health (Boucher et al., 2013; Rosenfeld et al., 2014ab; Seinfeld et al., 2016;
Jickells et al., 2005; Yu et al., 2015; Lim et al., 2012). These particles originate from both
human activity and natural processes, and they can cover vast regions of the globe.
Observations from satellite sensors provide the best means for monitoring and
quantifying the extent and transport of large-scale aerosol events (Kaufman et al., 2005;
Yu et al., 2012), and provide some characterization of aerosol particle properties (Remer
et al., 2005; Torres et al., 2013; Kalashnikova and Kahn, 2006; Kahn and Gaitley 2015).
Especially since the launch of NASA's Earth Observing System (EOS) and similar
satellites by international agencies, the community has benefitted from nearly two
decades of quantitative measures of the global aerosol system. While both passive and
active sensors have contributed to our understanding of the global aerosol system, here
we focus on only passive sensors. These, such as the MODerate resolution Imaging
Spectroradiometer (MODIS) (Levy et al., 2013; Hsu et al. 2013; Lyapustin et al., 2011),
the Multiangle Imaging Spectro-Radiometer (MISR) (Diner et al., 1998; Martonchik
1998; Kahn et al., 2010), the Ozone Monitoring Instrument (OMI) (Torres et al., 2013),
and POLarization and Directionality of the Earth's Reflectances (POLDER) (Tanré et al.,
2011) have provided instantaneous measures of aerosol loading, particle size, particle
absorption and aerosol type across the globe. The community has used these data to
calculate decadal statistics of aerosol climatology, seasonal and monthly statistics,
quantitative measures of intercontinental aerosol transport and fertilization of ecosystems
(Remer et al., 2008; Yu et al., 2012, 2013, 2015). These satellite aerosol products have
been used to estimate aerosol radiative effects and climate forcing, associations between
aerosols and cloud micro- and macrophysics, precipitation, air quality and public health,
and have provided critical constraints on global climate modeling (Zhang et al., 2005;
Koren et al. 2005; Lin et al. 2006; Wang and Christopher, 2003; Quaas et al., 2009;
Patadia et al., 2008; to give just one early example of each application).

The sensors mentioned above all have been launched on polar orbiting satellites in low
earth orbit (LEO). Such satellites are sun synchronous, passing over each location on
Earth at approximately the same local solar time each day. A LEO sensor imaging a
broad swath of Earth will image every spot on Earth, and thus achieve full global
coverage in 1 or 2 days. However, each spot on Earth is only viewed once per day in
daylight and once per day at night, always at approximately the same local solar time.  In
contrast a geosynchronous (GEO) satellite orbits at a high altitude above Earth, matching
the same period as the Earth's rotation. A sensor on a GEO satellite can scan the full or
partial portion of Earth's face presenting to the satellite. Neither the sensor nor the Earth
appear to move in these images although the terminator between day and night on the
Earth appears to move from east to west across the image over time. A GEO imager
always views the same Earth locations across approximately 1/3 of the Earth, and cannot
by itself provide full global coverage. However, a sensor on a GEO satellite can provide
information on the aerosol in any viewed location as a function of time of the day,
enabling monitoring of the diurnal cycle.

For about a decade there has been a publicly available operational aerosol product
derived from a GEO sensor. This is the GOES Aerosol Smoke Product (GASP) (Prados
et al.,2007), where GOES stands for Geostationary Operational Environmental Satellite.
GASP provides aerosol optical depth for the daylight section of the continental United
States at 4 km spatial resolution every 30 minutes in near real time, and the data is
archived. The sensor has only five channels, one spectrally broad channel (0.52-0.71 μm)
in the visible and four in the near to thermal infrared. The aerosol retrieval algorithm
makes use of the infrared channels for cloud masking, but must acquire all of its aerosol
information from a single visible channel. The lack of a channel in the shortwave infrared
(e.g. 2.1 or 2.2 μm) prohibits application of an EOS-era Dark Target retrieval (Kaufman
et al., 1997; Levy et al., 2007) and the lack of any channel in the blue eliminates the
possibility of a Deep Blue retrieval (Hsu et al. 2004, 2013). Thus the GASP retrieval is
handicapped by the relative primitiveness of the GOES-13 sensor. Even so, aerosol
optical depth (AOD) retrievals from GASP collocated and compared with AOD
measurements from the AERosol Robotic NETwork (AERONET; Holben et al. 1998) at
10 sites in the northeastern U.S. and Canada showed reasonable agreement. Regression of
GASP and AERONET AOD derived correlation of 0.79, rms difference of 0.13, and
absolute bias of less than 30% for larger AOD (e.g. AOD > 0.1). Validation in the
southeast and western U.S. was less good. The GASP validation statistics are reasonable,
but not as good as those produced by MODIS AOD retrievals at the same AERONET
locations. The main point of GASP, though, is not its absolute accuracy, but that it
provides quantitative information on the diurnal cycle of aerosol across the continental
United States and southern Canada.

We are now entering a new era in GEO observations. With the launch of the Advanced
Himawari Imager (AHI) (Yu and Wu, 2016) and the Advanced Baseline Imager (ABI) on
GOES-16 and GOES-17 (Kalluri et al., 2018, Kondragunta et al., 2019), we have sensors
in GEO orbit with spectral capability similar to MODIS. AHI has 16 bands, including
three in the visible and another in the SWIR. ABI also has 16 bands, but distributed
differently across the visible to the SWIR. This spectral capability combined with
nominal spatial resolution 0.5 to 2 km creates opportunity for aerosol retrievals that can
advance beyond what GASP could produce. Aerosol algorithms developed and
implemented by the agencies responsible for the operations of the GEO satellites are or
will be produced operationally and made public. These include the Japanese
Meteorological Agency (JMA) for AHI on Himawari (Uesawa, 2016) and the National
Oceanographic and Atmospheric Agency (NOAA) for ABI on the GOES-R series. In
addition to these official operational products, other algorithms have been developed that
make use of the new generation of GEO observations for aerosol retrievals, especially for
AHI data (Sekiyama et al., 2015; Yumimoto et al., 2016; Lim et al., 2016, 2018ab; Zhang
et al., 2018; Yoshida et al., 2018;Yang et al., 2018; Shi et al., 2018; Yan et al., 2018;
Choi et al., 2019). Some of these alternative aerosol products are research algorithms for
specific purposes, while others could be of general interest and could be made public.

Because the capabilities of the new GEO sensors replicate the important spectral and
spatial capabilities of the MODIS sensors, the MODIS Dark Target (DT) algorithms over
land and ocean (Remer et al., 2005; Levy et al., 2010, 2015, 2018; Gupta et al., 2016) can
be applied to AHI or ABI observations with only minor adjustments. The expectation is
that the resulting aerosol product will match the original MODIS product in terms of
accuracy and availability (number of retrievals). More than providing just another
alternative aerosol product to the community, application of the traditional DT algorithm
to GEO sensors offers continuity with a nearly 20-year well-studied, validated, and
accepted aerosol product. The continuity of a DT AHI or ABI algorithm means that there
could be an accurate MODIS-like aerosol product that resolves the day time diurnal
cycle, providing a well-understood quantitative measure of aerosol loading at fine
temporal resolution at the large regional scale.

In this study we present the results of the first attempt at porting the MODIS DT aerosol
algorithm to the AHI sensor on the Himawari-8 geosynchronous satellite. The study
makes use of a special limited data set of AHI spectral reflectances, prepared for research
purposes during the KORUS-AQ field campaign during May-June 2016. The purpose of
this work is to test whether there is any skill in applying the DT to AHI, and whether the
goal of a continuous time series of retrieved AOD from MODIS to AHI has any
probability of success. Furthermore, the study will identify issues that arise from the new
geometry, and demonstrate the ability of the new sensor to resolve aerosol signals using
the DT algorithm that previous sensors could not.

The AHI inputs and the algorithm will be described in Section 2, with emphasis made on
how the AHI algorithm differs compared to the MODIS implementation. Section 3 will
present results and compare these with standard MODIS retrievals and collocated
ground-based observations. Section 4 will explore the AHI aerosol product's diurnal
cycle at AERONET stations for validation, and then question how well the diurnal mean
AOD inferred from once-a-day LEO observations compare with a truer diurnal mean
compared from observations made at finer temporal resolution.  Finally, results will be
summarized and discussed in Section 5.

**2.0 Data and retrieval algorithm**

**2.1 AHI sensor characteristics**

The AHI was first launched on the Himawari-8 satellite in 2014 and became operational
in July 2015. It is in geosynchronous orbit over the equator at 140.7° East. The second
AHI was launched on the Himawari-9 in November 2016 and remains in a standby mode.
The instrument has the capability to image a mesoscale region every 30 seconds while
providing full disk coverage every 10 minutes. In this study, the full disk data have been
used. The data to be presented here were obtained from the University of Wisconsin
atmospheric Science Investigator lead Processing System (SIPS) which processed the
NOAA's operational cloud operating system – extended (CLAVR-x) which provides
radiance values at all 16 channels at a consistent 2km resolution as a
diagnostic/byproduct of the cloud retrieval. SIPS made the AHI data available
specifically to support the KORUS-AQ campaign and for research purposes, and thus
only two months of data were available. For this analysis we processed the DT algorithm
at 1-hour temporal resolution from 0000 UTC to 0800 UTC, 9 full disk images per day.
Figure 1 shows an example of the AHI full disk image. AHI wavelengths used in the DT
aerosol retrieval along with their spatial resolution are shown in Table 1, and compared
with their counterparts from the MODIS and Visible InfraRed Imaging Radiometer Suite
(VIIRS) instruments. From Table 1 we see that AHI nearly matches MODIS and VIIRS,
wavelength by wavelength in the bands needed by the DT algorithm, except for missing
the 1.24 µm band that is used in the aerosol retrieval over ocean and also in masking
snow/ice over land and sediments in the ocean. It is also missing the 1.38 µm channel
that the DT aerosol algorithm has relied on for identifying and masking thin cirrus. For
the bands that overlap MODIS, although close in spectral resolution, they do not exactly
match. For this reason the algorithm Look Up Tables (LUT), gas absorption corrections
etc. cannot be applied directly from the current MODIS algorithm and must be calculated
specifically for AHI.

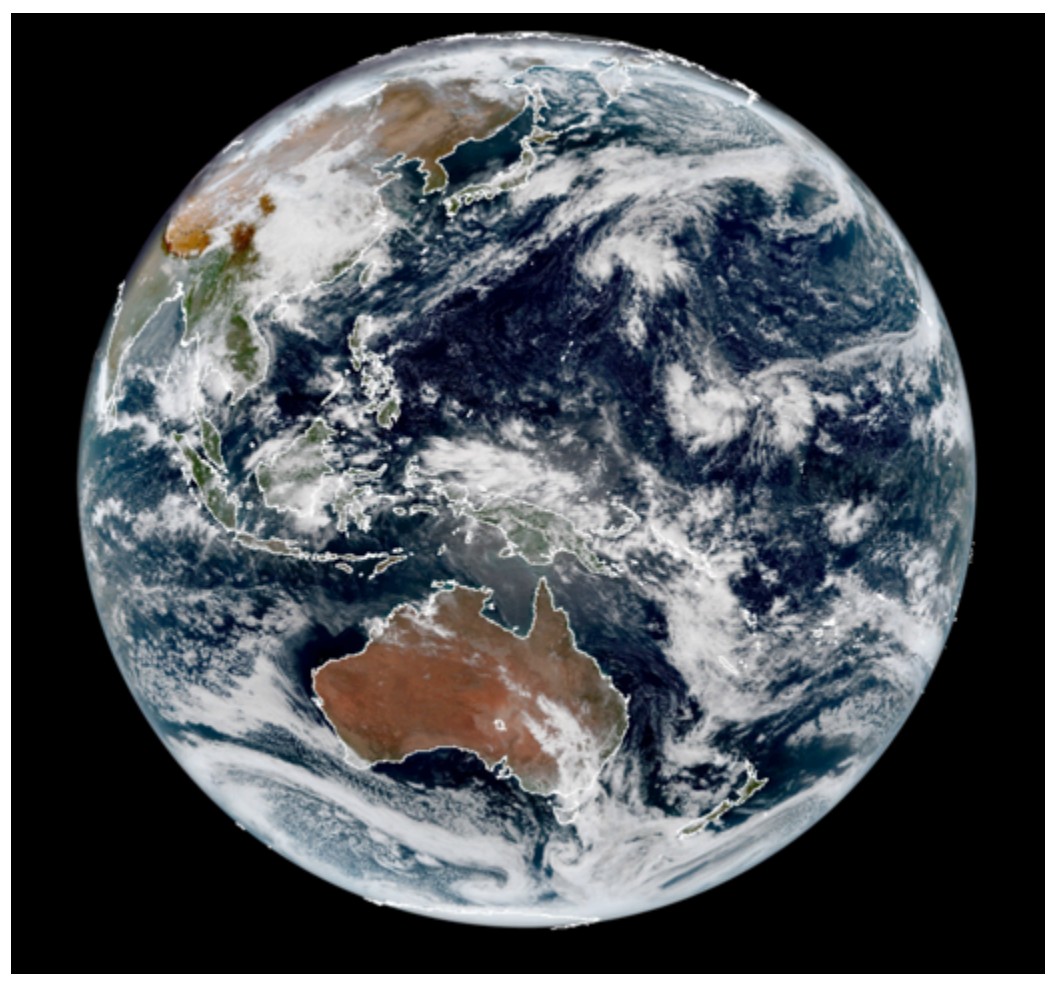


Figure 1. Full disk true color image from AHI using the 0.64 μm, 0.51 μm, 0.47 μm
channels. The image is taken on October 20, 2018 at 02:10 UTC.

AHI's native spatial resolution is coarser than MODIS's, but comparable to VIIRS. Note,
however, that the spatial resolution noted in Table 1 refers to the subsatellite point. The
spatial resolution of Earth scenes at the edges of MODIS and VIIRS swaths or at the edge
of the AHI disk will have spread from their subsatellite value. MODIS pixels spread by 4
times their nadir value, and VIIRS pixels spread by 2 times. AHI pixels remain 1-3 times
their size at the subsatellite point for all but the extreme edge of the full disk image. Also
note that the actual KORUS-AQ data used in this study have reduced spatial resolution in
all channels (2 km).



Table 1. MODIS, VIIRS and AHI wavelengths in μm used directly in the DT algorithm (**bold**) and subsatellite point spatial resolution in kilometers. The table presents native resolution of sensors, but this study uses a special run of AHI where all spectral channels were reduced to a resolution of 2 km.

| MODIS | VIIRS | AHI |
|---|---|---|
| **0.47**/0.5 | **0.49**/0.75 | **0.47**/1.0 |
| **0.55**/0.5 | **0.55**/0.75 | **0.51**/1.0 |
| **0.66**/0.25 | **0.67**/0.75 | **0.64**/0.5 |
| **0.86**/0.25 | **0.86**/0.75 | **0.86**/1.0 |
| **1.24**/0.5 | **1.24**/0.75 | |
| **1.38**/0.5 | **1.38**/0.75 | |
| **1.61**/0.5 | **1.61**/0.75 | **1.61**/2.0 |
| **2.11**/0.5 | **2.25**/0.75 | **2.25**/2.0 |

Given that the wavelengths and spatial resolution of AHI differ from the heritage DT aerosol algorithm means that while the structure, heritage and experience of the MODIS DT algorithm can be adapted for AHI to maintain as much continuity as possible, the resulting AHI algorithm and product will not be an identical twin.

**2.2     Dark Target AHI aerosol algorithm and research product**

The DT aerosol algorithms are a family of algorithms, based on the original two algorithms that retrieved aerosol over ocean and over land from the MODIS instruments aboard NASA's Terra and Aqua satellites. Levy et al. (2010, 2015, 2018) and the on-line Algorithm Theoretical Basis Document (https://darktarget.gsfc.nasa.gov) describe these algorithms in depth. Here we only provide an overview in order to highlight the differences between the original algorithms and the DT algorithm applied to AHI inputs. DT algorithms should not be confused with other operational NASA aerosol algorithms applied to MODIS inputs (e.g. Deep Blue; Hsu et al., 2013 and MAIAC: Lyapustin et al.,

2011).  Both DT ocean and DT land procedures use Lookup Tables (LUTs). LUTs are
created by using radiative transfer (RT) code to simulate spectral top-of-atmosphere
(TOA) reflectance for expected conditions of aerosols over a theoretical rough ocean
surface or black land surface. These LUTs assume intrinsic physical and optical
properties (size, shape, refractive index) as well as total column loading of atmospheric
aerosols.

The original DT retrieval relies on seven reflective solar bands for aerosol retrieval and
one for cirrus detection and masking (Table 1). Additional bands are used for tasks like
cloud masking, snow identification, etc. The algorithm adapted for AHI makes use of 6
bands for the aerosol retrieval that are similar, but not exactly the same as the original
MODIS ones. The differences require new corrections for trace gas absorption in the
channels, and the calculations of new LUTs. It is thus impossible to exactly duplicate the
DT algorithm as it is ported from sensor to sensor. However, the basic physical
assumptions, RT codes, algorithm architecture, and intrinsic physical, and optical
properties used to calculate the LUTs are the same in the AHI DT algorithm, as they are
in the current MODIS and VIIRS DT algorithms.

The greatest consequences to missing the 1.24 µm band is in sediment masking for ocean
and snow/ice masking for land. New techniques that compensate for the missing
information were applied to the AHI data. For sediment masking, we follow Li et al.
(2003) as is standard for the DT algorithm, but substitute the 1.61 channel for the
standard 1.24 µm channel. Physically this substitution should work, as both channels are
expected to be black in sea water, which provides the background from which sediments
are flagged, but the substitution has not yet been well-vetted. We have not yet devised a
substitute for the over land snow/ice mask. The data analyzed and shown here are from
May and June 2016, months when no snow is expected in the domain. Devising, testing
and implementing an AHI snow mask will be needed before the DT AHI algorithm can
be applied year round. In terms of the direct aerosol retrieval, the lack of the 1.24 µm
information only affects the over ocean algorithm slightly, as the information from the
0.86 µm and the two longer wavelengths compensate for its absence (Tanré et al., 1996;
1997).
The loss of the 1.38 µm channel may have more pronounced consequences as it proved to
be the first line of defense against thin cirrus contamination in the aerosol product (Gao
and Kaufman, 1995). In this initial adaptation of the algorithm to AHI we have not
implemented any alternative test for thin cirrus, and therefore cirrus contamination is
expected in the results shown here. For clouds other than thin cirrus, we apply an internal
cloud mask to the input radiances, similar to the traditional MODIS aerosol cloud mask.
This mask is based on spatial variability of groupings of 3x3 input radiance pixels
(Martins et al., 2002), and the same thresholds were used. However, while the MODIS
aerosol cloud mask also incorporates specific tests from the standard MODIS cloud mask
(MOD/MYD35: Frey et al. 2008) that are based on thermal infrared channels, those
products are not available for AHI. No direct substitution is employed to compensate.
The absence of these specific external cloud mask tests will mostly affect high cold cloud
identification. Because alternative methods have not been developed for masking clouds,
and the alternative method for identifying sediments has not been vetted to the same
extent as the original MODIS DT masking techniques, the possibility of contamination
from these features affecting the aerosol retrievals is higher than expectations based on
the MODIS heritage.
The traditional MODIS DT algorithm aggregates 20 x 20 pixels at 0.5 km resolution to
form a "retrieval box". These 400 pixels are screened for clouds, glint, sediments,
improper land surfaces and other elements. Then the remaining pixels that have escaped
the masking are sorted from high to low reflectance, and the darkest and brightest "good"
pixels are arbitrarily eliminated. Darkest is defined as the darkest 20% over land and 25%
over ocean. Brightest is defined as the brightest 50% over land and 25% over ocean. At
that point, the spectral reflectance from those pixels that remain after the 2-tiered
elimination process are averaged to represent the mean spectral reflectance in the nominal
10 x 10 km$^2$ retrieval box. The algorithm proceeds with the inversion using that
representative spectral reflectance and produces one set of aerosol properties
representative of the retrieval box.

The AHI retrieval algorithm adapts this MODIS process for its coarser spatial resolution
by aggregating 10 x 10 pixels at 2.0 km resolution to create retrieval boxes that have
nominal resolution of 20 x 20 km$^2$ (at the subsatellite point). The same 2-tier elimination
process using modified cloud, sediment, glint etc. masking, and removal of darkest and
brightest pixels is applied. Both the MODIS and AHI remove the same percentage of
dark and bright pixels. Because AHI starts the process with 100 pixels but MODIS with
400 pixels, there are fewer pixels to remove with AHI, and smaller number of pixels
remaining to be used to represent the spectral reflectance in the box with AHI. After the
representative reflectances have been calculated there are corrections for gas absorption
($H_2O$, $O_3$, $CO_2$). The result is a single set of spectral reflectances in the six bands that is
input to the retrieval algorithm. Additional inputs include ancillary data such as ozone
profiles, wind speed and water vapor columns from NOAA's Global Data Assimilation
System (GDAS) reanalysis data, and a global land/sea mask generated by CLAVR-x at 2
km resolution.

Whether ocean or land, the DT retrieval searches the pre-computed LUTs to find the best
match to the spectral observations. The over land algorithm makes use of measured
reflectance at 0.47, 0.66 and 2.1 μm and assumptions about the surface reflectance to
determine the aerosol loading and establish the relative weights between two aerosol
models, both defined by geographical location and season. Over ocean, the algorithm
uses six wavelengths (0.55, 0.66, 0.86, 1.24, 1.61 and 2.13 μm) to determine the aerosol
loading and define an aerosol model from one fine mode and one coarse mode, and the
relative weight between these modes. There are no restrictions on the distribution of
modes by location and season in the ocean algorithm. Once the aerosol model is defined
by the weighting between models or modes, the spectral extinction of the aerosol is
defined. The retrieved aerosol loading can be translated to AOD at any wavelength
because of the known spectral extinction, and all wavelengths are reported in the output.
The primary wavelength we will use here is AOD at 0.55 μm. Two measures of aerosol
particle size are given for the over ocean retrieval, Fine Mode Weighting and Ångström
Exponent (AE). AHI-retrieved aerosol size parameters will not be examined in this paper.
Although the ocean and land retrievals have similarities, the details are different because
land surface optical properties are different than ocean. The ocean algorithm calculates a
"rough" surface (whitecaps, foam, glitter), which is a function of wind-speed, while the
land algorithm assumes quasi-static ratios between blue (0.47 μm), red (0.64 μm), and
shortwave infrared (e.g. 2.25 μm) wavelengths. Land surface ratios for the retrievals
shown in this study are identical to those used by the standard MODIS Collection 6.1
algorithm. Different wavelengths and different viewing geometry may introduce
unexpected uncertainties. Of particular concern is the assumption that LEO land surface
ratios will hold for the new GEO view geometry. Previously land surface ratios were
found to have only a weak dependence on the viewing geometry encountered by a LEO
observation (Levy et al., 2007), but the range of geometries encountered by a GEO
instrument are different and require further analysis. Still, the original assumption of
predictable surface reflectance ratios is based on the physical linkage between
chlorophyll and liquid water light absorption that should continue to transcend
Bidirectional Reflectance Distribution Function (BRDF) and other angular effects. In
addition to the aerosol properties, DT provides many diagnostics including Quality
Assurance and Confidence (QAC).

The new AHI DT algorithm was applied to input AHI full disk radiances, daylight
portion of the disk-only. View angles were confined to less than 72 degrees and solar
zenith angles were restricted to less than 80 degrees. The period of analysis spans two
months (May-June 2016). Given 9 images per day, the data base for analysis thus
includes more than 549 disk images of AOD derived from AHI inputs using the new AHI
DT algorithm.

**2.3 MODIS aerosol products**

The AOD retrieved from AHI using the DT retrieval will be compared with the more
established and well-characterized DT AOD product from MODIS on board the Terra
and Aqua satellites. Specifically we will be accessing Collection 6.1 Level 2 MOD04 and
MYD04 data products, where MOD refers to products derived from Terra MODIS inputs
and MYD refers to those derived from Aqua MODIS inputs. Level 2 refers to derived
geophysical parameters from the Level 1b geolocated and calibrated measured radiance
inputs. Level 2 data are provided in 5 minute cut sections of the orbital image called
granules. These images are not gridded, but instead follow directly from the instrument
scan as it follows its orbital path. There are many individual aerosol and diagnostic
parameters within each MOD/MYD04 file. This study makes use of only one parameter,
Optical_Depth_Land_And_Ocean. This parameter combines the retrieved AOD at 0.55
µm from the independent algorithms applied separately over land and ocean, and uses
only those retrievals identified with the highest quality (QAC=3 over land and QAC > 0
over ocean). MODIS granules were selected that fall within the daylight portion of the
AHI radiances, corresponding to the same days of the AHI images analyzed. Further
temporal (±0.5 hour) and spatial (0.25x0.25 degree) collocations have been performed for
specific analysis.

**2.4 AERONET aerosol products**

AERONET is a global ground network of CIMEL sun-sky radiometers and data processing
and analysis software commonly used to evaluate satellite-derived aerosol products
(Holben et al., 1998). In this work, only the direct sun measurements will be used.
AERONET processes these spectral measurements to derive AOD at the wavelengths
corresponding to the direct sun measurements. The AERONET spectral AOD product is a
community standard for satellite-derived AOD validation, given that AERONET's AOD
uncertainty of 0.01-0.02 (Eck et al., 1999) is sufficiently more accurate and precise than
can be expected by any satellite retrieval. The configuration of the spectral bands varies,
but typically is centered at 0.34, 0.38, 0.44, 0.50, 0.67, 0.87, and 1.02 µm. Here we use a
quadratic log-log fit (Eck et al., 1999) to interpolate AERONET AOD to 0.55 µm to match
the AHI AOD product. The typical temporal frequency of direct sun measurements is every
15 minutes. The network consists of hundreds of stations, located globally, across all
continents and in a wide variety of aerosol, meteorological and surface type conditions.
Here, we only include stations within the AHI view disk. AOD data from AERONET are
reported for three different quality levels: unscreened (level 1.0), cloud screened (level 1.5)
and cloud screened and quality assured (level 2.0). We will only use Version 3 Level 2.0
AERONET AODs in this study.
**3.0 Comparison with AERONET and MODIS DT**
The new AHI DT algorithm was applied to AHI-measured radiances over the full disk
(except for extreme viewing and solar angles), daily, through the 9 measurement times
(hourly: 0000 to 0800 UTC). We will test this new product by first validating it against
collocated AERONET measurements and then comparing it with the well-vetted MODIS
DT product.
**3.1 Validation against collocated AERONET AOD**
The validation procedure requires calculating the spatio-temporal statistics of a collocated
AHI-retrieved and AERONET-measured AOD pair (Ichoku et al. 2002; Petrenko et al.,
2012; Munchak et al., 2013; Remer et al., 2013, Gupta et al., 2018). Thus, the temporal
mean AOD of all AERONET AOD measurements within ±30 minutes of an AHI scan
will be compared with the spatial mean of all Level 2 AHI-retrieved AOD values within a
0.25x0.25 degree box centered at the AERONET station. This method of matching
spatio-temporal statistics, in one form or another, has become a standard within the
aerosol remote sensing community (Levy et al., 2010; Petrenko et al., 2012; Remer et al.,
2013, Huang et al., 2016, Gupta et al., 2018). As new satellite aerosol product types have
been introduced, the specifics of the spatio-temporal match-ups have been re-evaluated.
For example for the DT MODIS 3 km product different temporal and spatial averaging
windows were investigated, with smaller windows chosen to better test the ability of the
finer resolution product to capture spatial gradients at less than 10 km scales (Remer et
al., 2013). As the DT geosynchronous products mature, we will conduct a similar
investigation into better ways to validate the ability of the new products to represent the
immediate diurnal cycle of the AOD at an AERONET station. For now, our purpose is to
see if the product from ported the algorithm can match AERONET at a very basic level,
and we will use the standard match-up procedure at traditional scales. The validation
exercise with AERONET only considers AHI AODs pairs with highest quality AHI
retrievals.

From the collection of these ordered pairs of collocated AHI and AERONET AODs a set
of correlation and regression statistics will be calculated, assuming that the AERONET
AODs are the independent variables and the AHI AODs are the dependent variables.
These include number of AOD pairs (N), the correlation coefficient (R), the slope (m)
and intercept (I) of the linear regression through the points, the overall mean bias and
Root Mean Square Error (RMSE) of the AHI AODs. Also we apply the expected error
(EE), based on previous validation of MODIS DT AODs against collocated AERONET
(Levy et al., 2013). We show the percentage of AHI AODs that fall within the EE
bounds. EE gives us a sense of whether a new product is meeting the standards of the
original product, which in itself has become a standard within the aerosol remote sensing
community. Another metric that could be used would be the Global Climate Observing
System (GCOS) criteria for AOD, which is 0.03 or 10%. This is a more stringent
requirement than what we have been able to achieve with the DT algorithm applied to
MODIS for 20 years, or to VIIRS. Thus, the GCOS requirement is not shown on the
validation plots, as it is certainty out of reach for this first test of DT applied to a GEO
sensor.

Figure 2 shows the results of this validation for the over land retrieval, with Fig. 2a
showing the scatterplot of the accumulation of all collocations for the duration of the time
period investigated, and also specific panels showing the same, but for individual
AERONET sites. The specific stations were chosen to represent three different validation
situations: when DT is biased high, biased low and unbiased against AERONET. Figure
3 shows the validation statistics calculated for each AERONET location within the AHI
domain. Altogether there were 1982 collocations during the period of the study, with a
dynamic range spanning AERONET-measured AOD from less than 0.05 to nearly 2. The
AHI AODs match AERONET observations with a correlation coefficient of 0.84, a mean
bias of 0.09 and RMSE of 0.20. Approximately 55% of the retrievals fall within the EE
that was based on MODIS validation. Figure 3 shows that the distribution of validation
statistics varies from station to station, but that correlations tend to be overall high across
mainland Asia, while biases, RMSE and percent within MODIS DT Expected Error vary
more widely, even within tightly packed local networks. The variability in AHI AOD
performance against AERONET over the domains comes from various reasons, including
variations in surface reflectance characterization (i.e. different type of land use type),
variability in assumed aerosols models within the algorithm and availability of high
quality valid AOD retrievals over individual stations. Often AOD is biased high when
surface reflectance ratios do not conform to assumptions. Such was the case for many
years with urban surfaces, until Collection 6.1 made an alteration (Gupta et al. 2016).
Even with that alteration, DT retrievals over Beijing continue to be high (Figure 4). Low
biases will occur when the assumed aerosol model is underrepresenting the amount of
light absorption of the particles. The land aerosol model used in this region in this season
is the moderately absorbing aerosol in May and the non-absorbing model in June. If the
aerosols are actually absorbing in June or more heavily absorbing in May in a particular
locality, such as at KORUS_UNIST_Ulsan, then   the retrieved AOD will be biased low.
The DT algorithm is designed for global-scale representation of the aerosol system,
which for GEO means full disk retrievals. The goal is to provide the most accurate
retrieval at each individual location, but the reality is that on the global scale we cannot
fine-tune land surface and aerosol model assumptions for each individual location, and
some locations will have products that are biased high and some biased low. The
difficulty in matching AERONET at individual stations is one of the limitations of the
current DT algorithm.

As a comparison, Figure 4 shows a similar set of plots, but for MODIS DT retrievals
against AERONET. These collocations were made at the same stations as in Fig. 2, and
over the same time period. Both Terra and Aqua are included. First, we see about half as
many points as were seen in the AHI collocations because Terra and Aqua MODIS each
pass over the area only once per day during daylight hours, while AHI is scanning these
sites up to 9 times per day. Second, we notice that MODIS AODs match collocated
AERONET AODs about the same as AHI AODs with R = 0.91, bias = 0.10, RMSE =
0.19, and with 55% within EE. Only the correlation between MODIS and AERONET is
substantially better than between AHI and AERONET.

We see from this limited validation that the AHI-retrieved AOD is sufficiently accurate
to represent the aerosol in this region, during this time period, approaching the same
validation statistics as the durable MODIS product. We note here that approaching the
same validation statistics as the MODIS product will still fall short of the more stringent
GCOS criteria. Additional analysis of AHI AOD biases as a function of surface
reflectance, aerosol typing, season, and sensor and satellite geometry required data
covering a longer time period. We plan to perform a more robust analysis in our ongoing
and future research before making the product operational.  We will next compare the full
overlap of AHI-retrieved AOD with MODIS retrievals, regardless of AERONET.





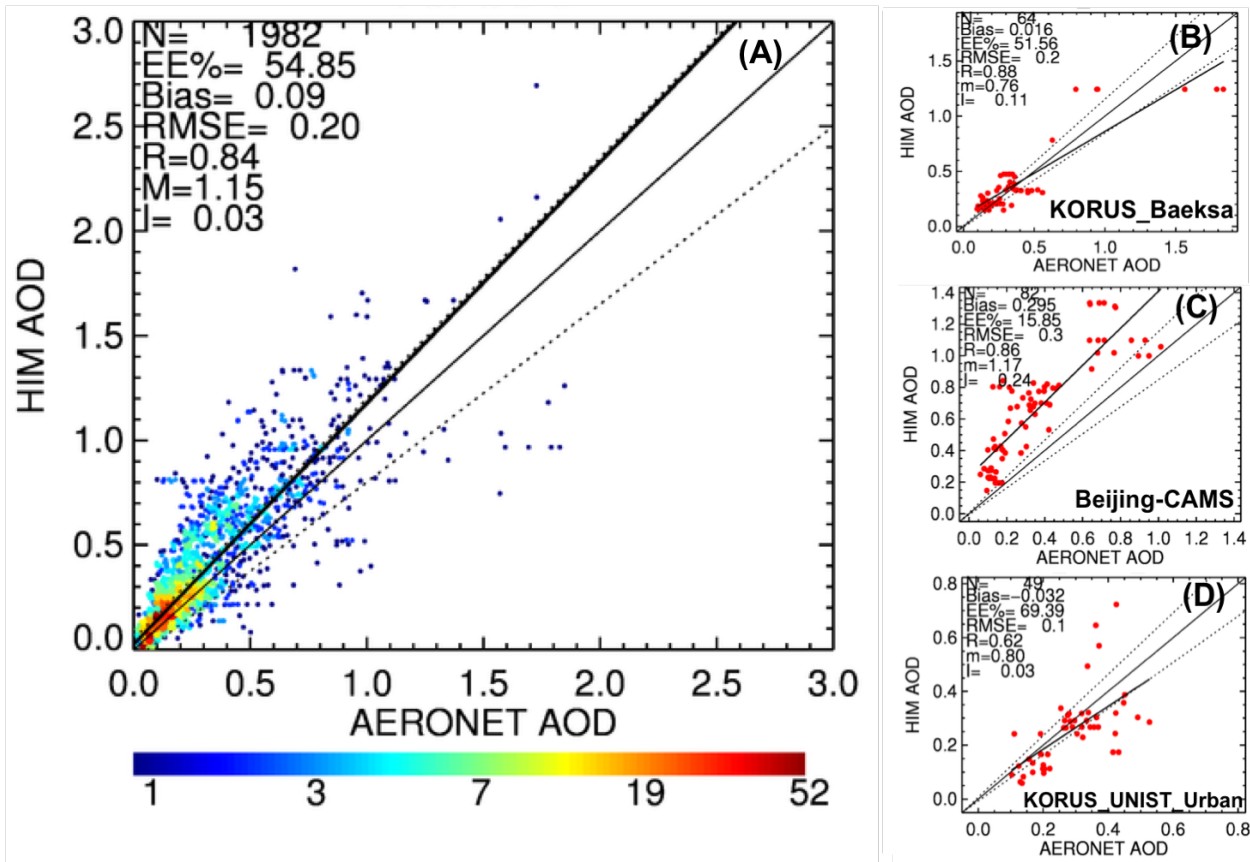


Figure 2. Density scatterplots of retrieved AOD at 550 nm derived from AHI radiances
using the new DT AHI algorithm versus AOD at 550 nm spectrally interpolated from
measured AODs from AERONET instruments collocated in time and space. Left panel
A. All accumulated collocations in the AHI full disk domain over the 2 months study
period May-June 2016. Right panels B, C and D, same for individual stations KORUS-
Baeksa and KORUS-UNIST_Ulsan in Korea and Beijing-CAMS in China. Shown in
each panel are the number of collocations (N), percent within expected error as
determined from MODIS DT analysis (EE%), mean bias (Bias), Root Mean Square Error
(RMSE), correlation coefficient (R), slope (m) and intercept (I) of a linear regression
equation through the points.

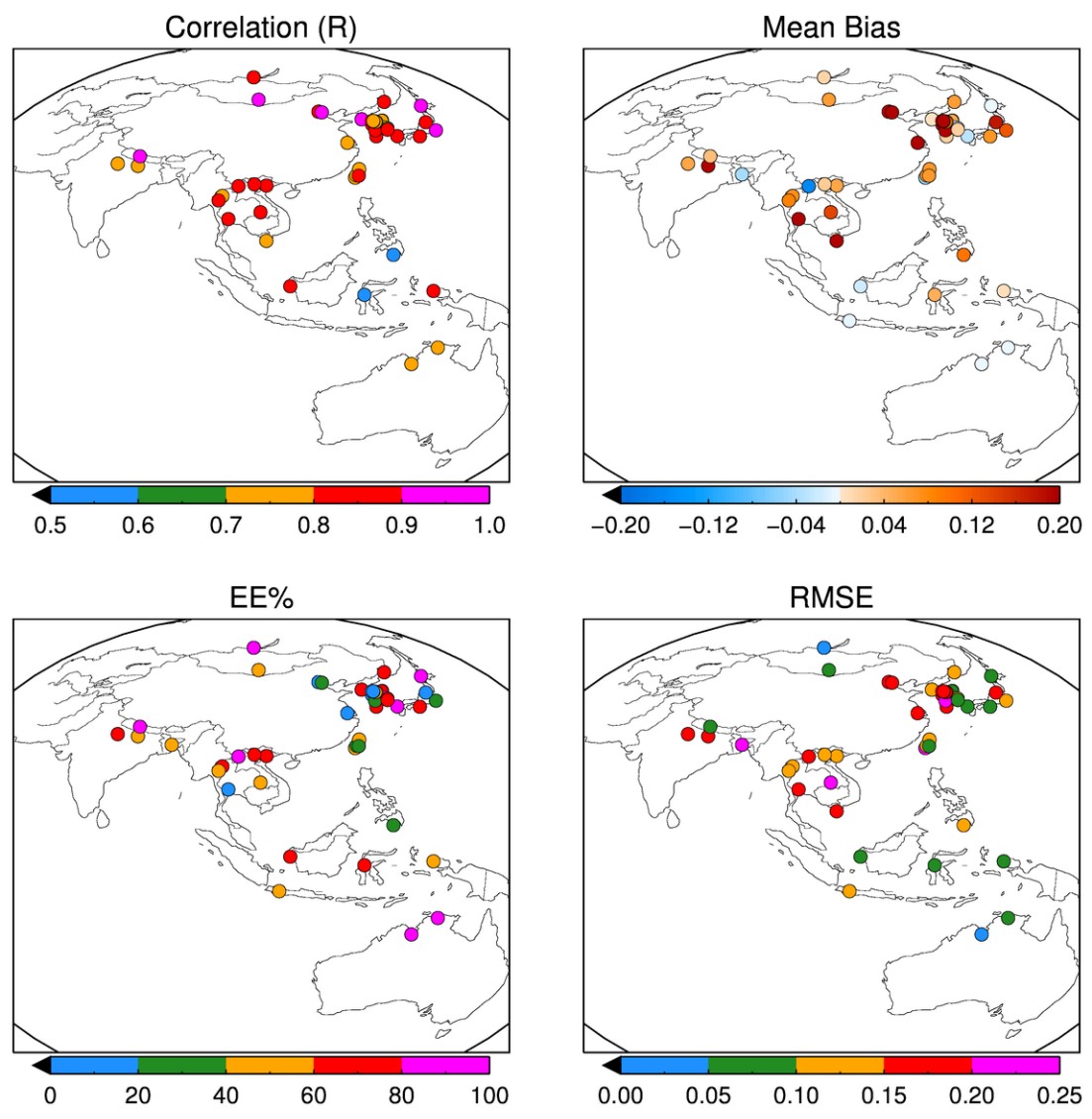



Figure 3. Spatial distribution of the collocation statistics between retrieved AOD at 550

nm derived from AHI radiances using the new DT AHI algorithm versus AOD at 550 nm

spectrally interpolated from measured AODs from AERONET instruments collocated in

time and space. Shown are upper left: correlation (R); upper right: mean bias; lower left:

Percentage within expected error (EE%); lower right: RMSE. Each point represents an

AERONET station location.


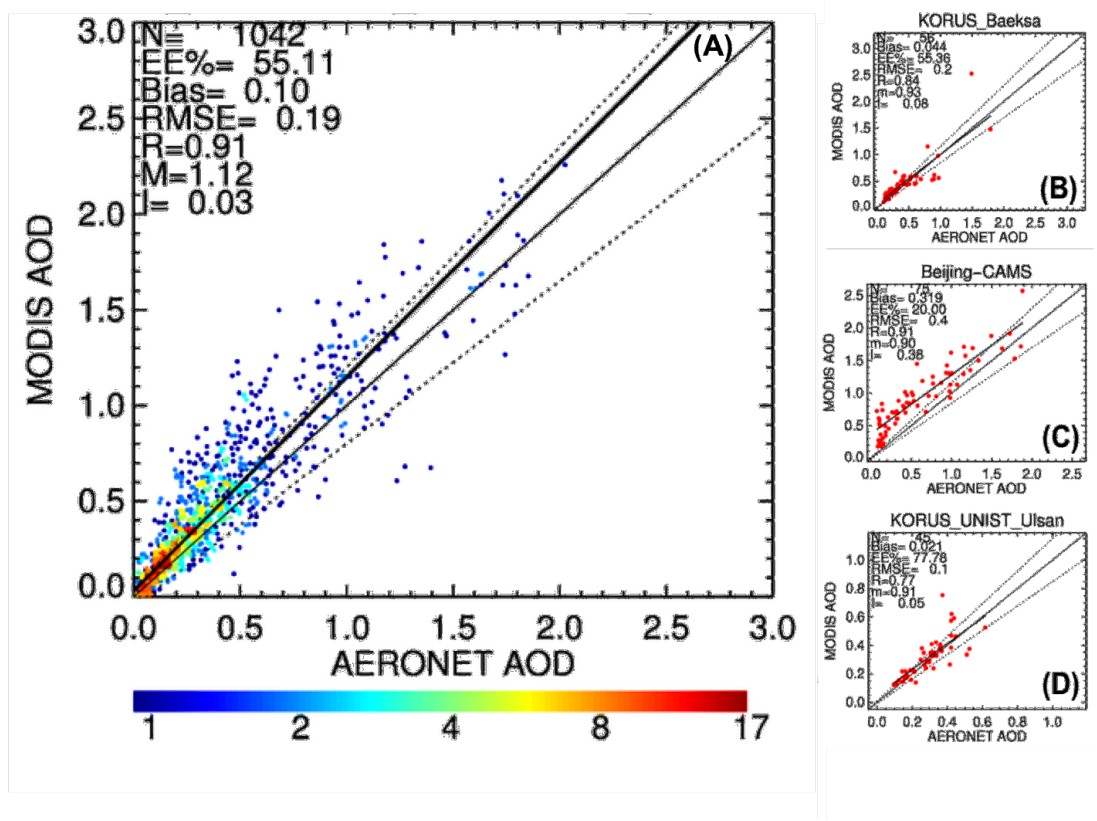


Figure 4. Same as Figure 2 except now density scatterplots of retrieved AOD at 550 nm
derived from Terra and Aqua MODIS using the operational DT Collection 6.1 algorithm.
This data represents same stations and time period as shown in Fig2.

**3.2 AHI versus MODIS DT**

To collocate AHI and MODIS AOD, the Level 2 MODIS and AHI AOD data were
mapped to a common 0.25° latitude by 0.25° longitude grid for each AHI full disk scan.
To fill the grid, we include all MODIS retrievals within ±30 minutes of the AHI scan. All
the AOD retrievals falling within the above spatial and temporal windows were averaged
and statistics are retained for further analysis. It takes MODIS approximately 35-45
minutes to cut a poleward-to-poleward swath across an AHI image, and about 6-7 swaths
to transverse east-west across the disk. Thus, in the common grid, at any particular time,
while most of the grid has the possibility to include an AHI retrieval when cloud and glint
free, only a relatively small portion of the grid will be filled with MODIS retrievals to
create the possibility of a collocation.
Figure 5 presents the scatter plots from matching the products of the Terra/Aqua and AHI
sensors on the common grid in each subset. Terra and Aqua collocations are kept
separate, as are over land and over ocean retrievals. The DT AHI-retrieved AOD and the
DT MODIS-retrieved AOD exhibit excellent correlation and similarity, as is expected
from applying nearly the same retrieval algorithm to the radiance measurements of both
sensors. Over ocean there are over 600,000 match-ups for Terra and over 1 million for
Aqua. The geosynchronous AHI retrievals match the polar-orbiting MODIS retrievals
with essentially zero bias and RMSE of 0.05 or less. Correlation between the two data
sets is 0.93 or greater. Over land, there are over 100,000 match-ups for each satellite with
no bias for Terra and 0.02 for Aqua, and RMSE of 0.09 or less. Correlations exceed 0.95
over dynamic ranges from 0.0 to approximately 2.0. The plots in Figure 5 show how well
the new AHI-retrieved AOD matches its MODIS counterpart when both AHI and
MODIS offer retrievals for a particular time and location. These plots do not address
situations in which a retrieval occurs for one satellite, but not the other, and therefore do
not address typical retrieval issues such as cloud masking and choosing appropriate
situations for the DT algorithm to make a retrieval. There can be also differences in
AODs from two sensors due to difference in their viewing geometries. This is something
beyond the scope of this paper and will be addressed in subsequent research.

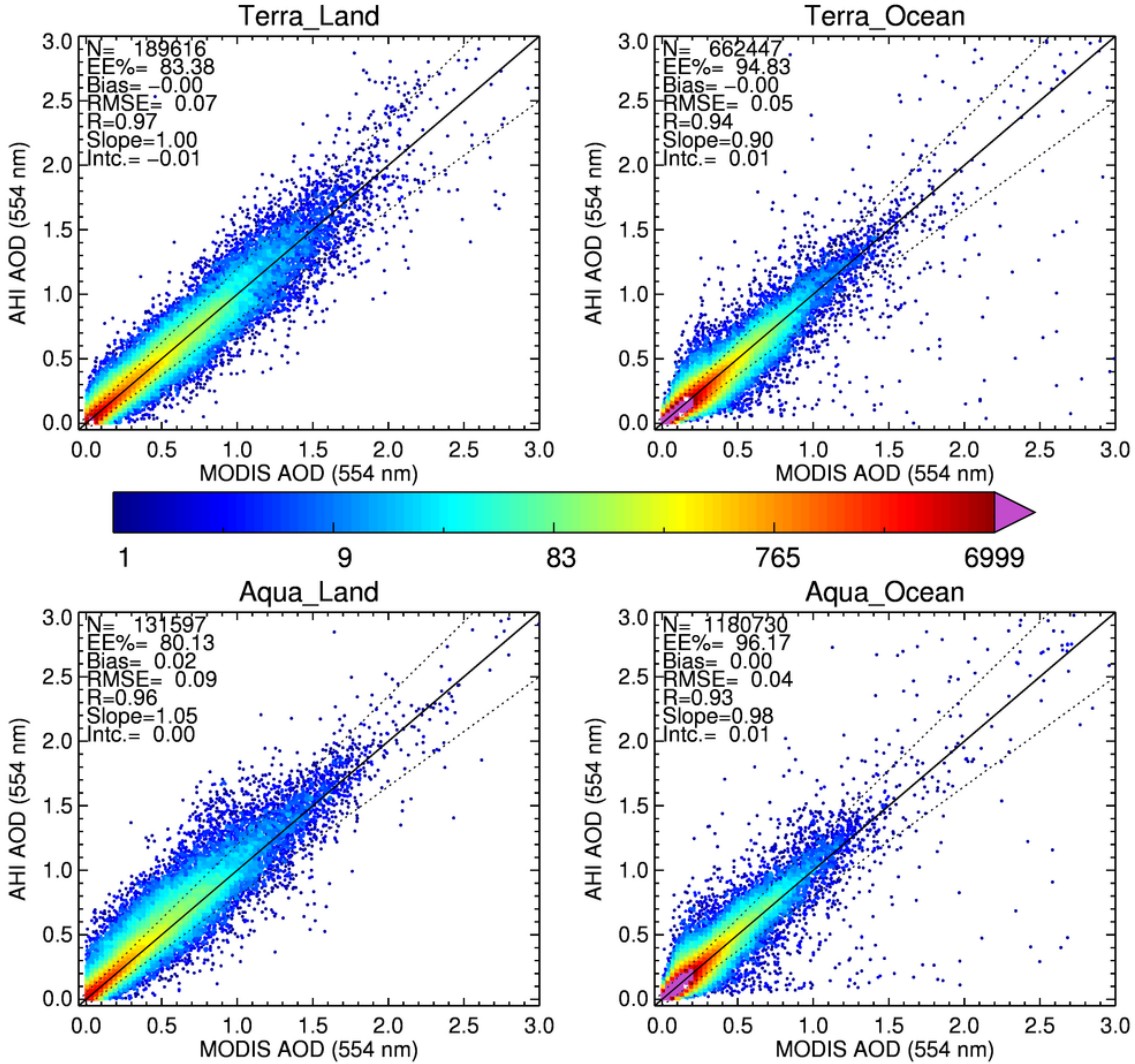

Figure 5. Density scatterplots of retrieved AOD at 554 nm derived from AHI radiances using the new DT AHI algorithm versus retrieved AOD at 554 nm from the operational MODIS Collection 6.1 algorithm, collocated in time and space. Top row are Terra MODIS. Bottom row are Aqua MODIS. Left panels are results from the over land retrieval. Right panels are results from the over ocean retrieval. The same collocation statistics are displayed as in Figure 2.

Figure 6 shows the two-month mean AOD over the AHI disk from AHI and Aqua MODIS, calculated from the data mapped to the common grid during our study period.

The mean AODs plotted here are collocated, and represent the AHI-derived AOD at
approximately the same time as the MODIS overpass.

We see that the DT algorithm applied to both sensors results in very similar distributions
of mean AOD across the AHI full disk image (Figure 6). This is despite the different
sensor characteristics and very different viewing geometries. There is elevated aerosol
across south and southeast Asia, and a separate pocket of elevated aerosol in northeast
China. Low AOD occurs across most of the tropical and southern oceans. Australia is
very clean and both sensors show a bit of moderately elevated AODs over the Indonesian
island of Java. The magnitude of mean AOD in these images ranges from near 0.0 to
almost 1.0.

The bottom panels of Figure 6 show the absolute differences in AOD when subtracting
the top row MODIS panel from the top row AHI panel, and a similar difference map
showing the differences between the top panel AHI values and a similar MODIS plot, but
from the Terra satellite. The difference maps are AHI minus MODIS so that positive
values, in red, indicate that AHI is higher than MODIS, while the negative values, in
blue, indicate that AHI is lower than MODIS. The range of differences span +0.10 to
about -0.08.

These plots indicate that over the elevated AOD regions, AHI retrievals are higher than
MODIS retrievals by as much as 0.10. This higher AHI AOD is more prevalent and
widespread with MODIS Aqua than with MODIS Terra. AHI tends to be about 0.02 to
0.03 higher than MODIS Aqua over much of the ocean regions surrounding the Asian
and maritime continents, while AHI tends to be closer to MODIS Terra in these regions
and sometimes even negative. Over Australia, AHI is less than MODIS Terra by as much
as -0.08. Because AOD values over Australia are very low to begin with, this negative
with respect to MODIS Terra indicates that AHI retrievals over Australia are often
absolutely negative more consistently than the MODIS retrievals and suggest that some
adjustment to the surface parameterization in the AHI DT retrieval will be required.

The inconsistencies between the two difference maps, one showing AHI with respect to
MODIS Aqua and the other with respect to MODIS Terra highlight the difficulty in
producing consistent representations of the AOD field, even when applying the same
algorithm to different sensors that should be exact duplicates of each other as in the case
of MODIS Terra and MODIS Aqua (Levy et al., 2018). Given this inconsistency between
the two MODIS instruments, the differences between AHI results and both MODIS
instruments fall within expected and manageable ranges. The DT algorithm as applied to
AHI is producing a representation of the spatial distribution of AOD with the same level
of fidelity as the original DT MODIS algorithm. This is the first attempt of applying the
DT algorithm to AHI, and we expect that future refinements to algorithm assumptions
that account for specific instrument characteristics and calibration will bring AHI AOD
results even closer to MODIS and AERONET values of AOD.


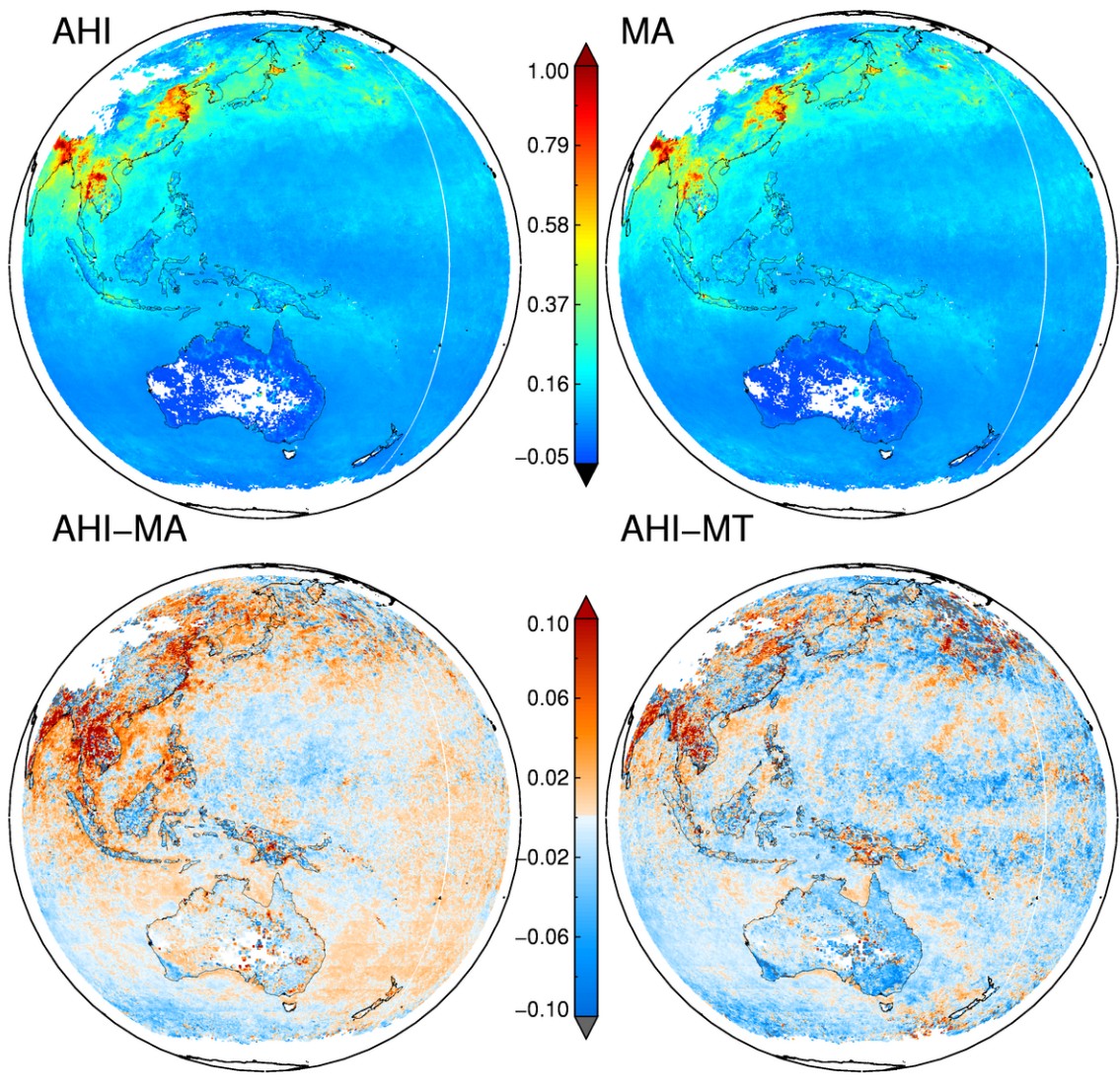



Figure 6. Top row: Mean AOD at 550 nm over the 2-month study period of May-June
2016. Upper left: mean AOD derived from retrievals using the new DT AHI algorithm
applied to AHI. Upper right: mean AOD derived from the standard Aqua MODIS DT
product. Bottom row: Difference maps of mean AOD at 550. Bottom left: Difference
between the two maps in the top row. Bottom right: Similar difference map but between
AOD from AHI and AOD from Terra MODIS (MT), instead of Aqua MODIS (MA).





## 4.0 Representation of AOD diurnal cycle using DT algorithm

### 4.1 Comparing AHI-derived diurnal signatures with AERONET

In the previous section we show how well the new DT AHI algorithm matches the AOD measurements from AERONET and the retrievals from MODIS. However, the point of applying an aerosol retrieval algorithm to a geosynchronous satellite sensor such as AHI is not to match the individual station data of AERONET nor the once-per-day retrievals from MODIS. The point of porting the DT aerosol algorithm to AHI is to represent the diurnal cycle of AOD over the broad regional area covered by the AHI full disk. In this section we explore the diurnal cycle of AOD derived from AHI and evaluate how much of the aerosol system MODIS has been missing because of its limited temporal sampling.

The diurnal cycle of AHI-derived AOD is compared with collocated diurnal patterns of AOD exhibited by AERONET stations within the AHI full disk image. The diurnal cycle at each AERONET station was calculated by finding the mean AERONET AOD at seven specific times of the day corresponding to the time of an AHI scan. These times are 01:00, 02:00, 03:00, 04:00, 05:00, 06:00 and 07:00 UTC, corresponding to the hours of 10:00 to 16:00 in local Korean time. All AERONET AOD measurements ±30 minutes of the nominal time were included in the average to represent the mean AOD at the nominal time. In parallel, the mean AOD at these specific times were calculated from all high quality AHI-derived level 2 AOD located within a 0.25x0.25 degree box centered around the AERONET station for all AHI scans taken at the nominal time. Thus, we created two representations of the diurnal cycle of AOD at each AERONET station, one from AERONET data and one from AHI-derived data, all from the collocation data set. This means that both AERONET and AHI must report at the same specific time for the instruments' AOD to be included in the calculated hourly average. This is the purest means to compare the actual retrieval, but will not reveal differences in sampling factors such as cloud masking because AHI will benefit from AERONET's cloud identification.

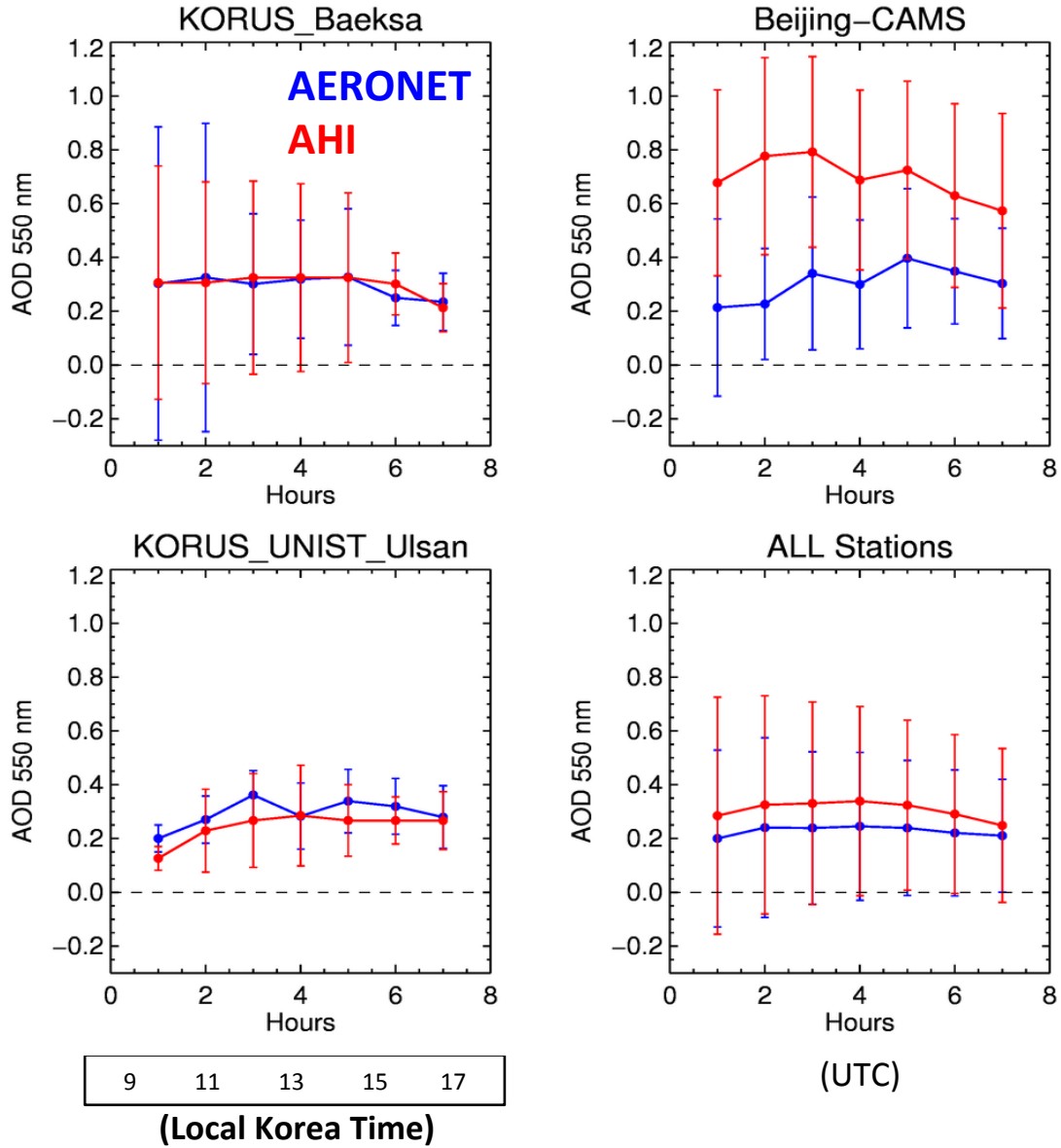


Figure 7. Median AOD at 550 nm in each of 7 time-of-day bins corresponding to 10:00 to
16:00 Korean standard time. Shown are three individual AERONET stations and also
(lower right) the results of binning all of the AERONET stations across the full disk
image as shown in Figure 3. Red indicates AOD derived from AHI using the new DT
algorithm. Blue indicates AOD measured by AERONET. The statistics were calculated
from the collocation data base such that each bin contains the same number of
observations from AHI and AERONET taken at the same time, although the number of
observations in each diurnal bin will differ. Vertical error bars represent one standard
deviation among different days for the same hour.
Figure 7 shows the calculated median AOD diurnal cycles from AERONET and AHI-
retrievals for three individual stations in Korea and China, and also the median of all
stations located in the AHI full disk image and reporting during our period of study. Error
bars represent the standard deviation of the sample in each hourly bin. At the three
stations shown individually in Figure 7, we see that the same biases seen in Figure 2 also
appear here, particularly with Beijing CAMS showing a strong positive bias. There is
wide scatter in the AOD for each hourly bin, as portrayed by the relatively large error
bars. The diurnal pattern of AOD, as measured by AERONET at KORUS Baeksa shows
a sudden decrease after 0500 UTC (14:00 Korea Standard Time), dropping from a steady
0.3 to 0.2 in two hours. The AHI AOD retrievals match this pattern almost exactly. The
other Korean station, KORUS_UNIST_Ulsan, shows an opposite daily pattern, with
AOD increasing from a morning low of 0.2 at 0100 UTC (10:00 Korean Standard Time)
to 0.3-0.4 at midday and then a drop off towards evening. The AHI AOD at this station is
biased low throughout most of the day, but does reflect the same diurnal signature of
increasing AOD over the morning. At the third station, Beijing CAMS, the AHI AOD
diurnal pattern does not match AERONET as well, but there is a strong positive bias
there with very large scatter in each hour. With error bars spanning 0.5 AOD, it is
difficult to discern diurnal changes with amplitudes of 0.2 AOD or less in either
AERONET or AHI.

The diurnal analysis shown in Figure 7 suffers from relatively small data samples. The
number of collocations for KORUS_Baeksa, KORUS_UNIST_Ulsan and
Beijing_CAMS are 56, 45 and 75, respectively, distributed over 7 hourly bins. If clouds
were not a factor, each hourly bin median might be constructed from only 6 to 11
samples. However, clouds are indeed a factor, with their own diurnal patterns. The actual
number of AHI-AERONET collocations at any particular hour might be as few as 3, and
morning and afternoon bins reported in Figure 7 might be constructed from entirely
different days. Therefore, the diurnal patterns in Figure 7 may be artificial composites
and not representative of the actual changes in AOD over the course of a single day.
However, the point of this comparison is not to speculate on the cause of the diurnal
signatures, but to establish that the AHI-derived AOD has the ability to describe the same
mean diurnal pattern in the aerosol as AERONET for individual locations.

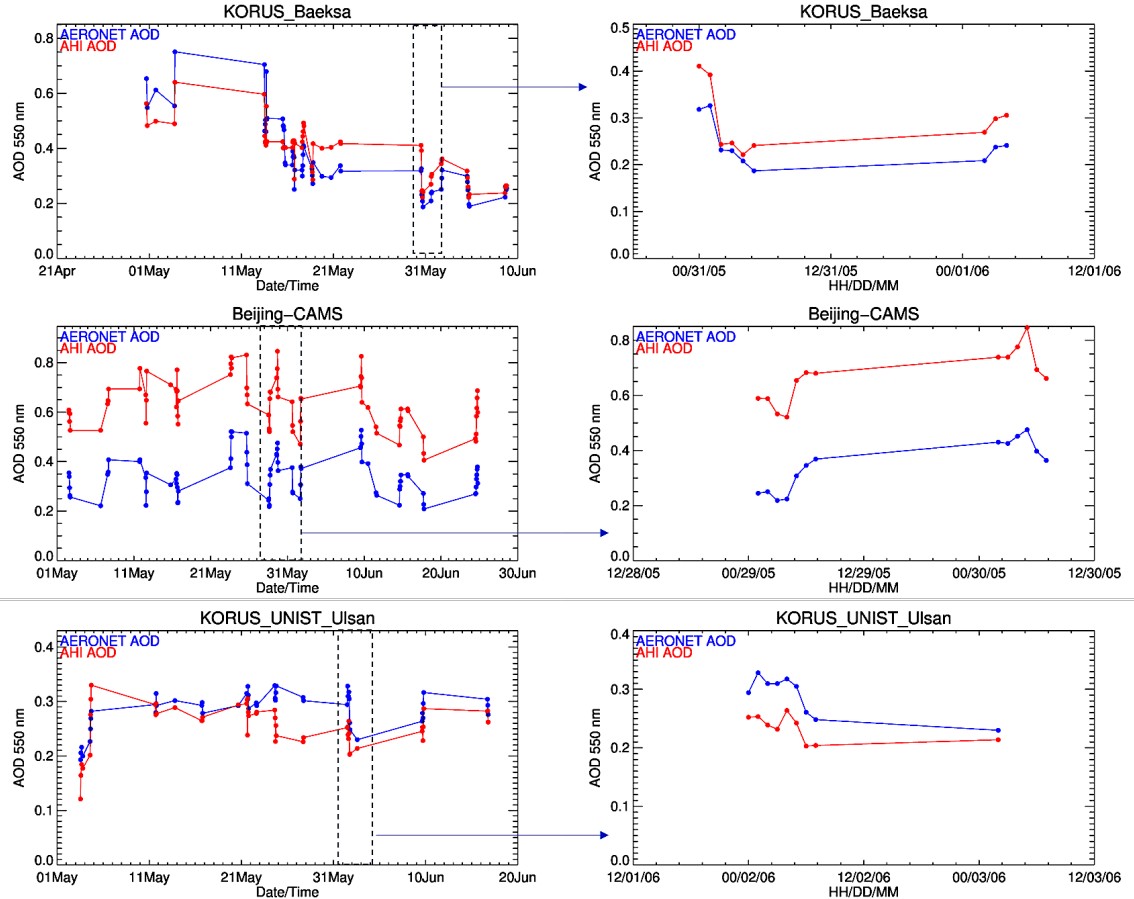


Figure 8. The time series of spatio-temporal mean AODs from AERONET (blue) and
AHI (red) for each hour of observation during the KORUS-AQ field campaign for the
same three stations as shown in Figure 7 (left panels). The right panels  zoom into
selected days as shown in the box with dotted lines in the left panels for each station.

Figure 8 further demonstrates the capability of AHI retrieved AODs to represents realistic
diurnal cycles over these three stations on individual days rather than in average sense as
shown in Figure 7. This analysis shows that AHI retrieved AODs follows AERONET
AODs hour-by-hour and day-by-day with apparent positive and negative biases over
different stations as discussed in the earlier section. Additional KORUS-AQ time series
of AHI and AERONET AOD for 46 other stations are shown in the Supplemental
Material. While there are some stations where AHI AOD does not follow the AERONET
temporal variability as well as those shown in Figure 8, most do.

The ensemble statistics of the diurnal signature for all AERONET stations and collocated
AHI retrievals in the AHI full disk image show the high bias of the AHI retrievals, as per
Figure 2, but also that the ensemble mean diurnal signature of AHI AOD is mostly flat, as
is the diurnal signature from AERONET. Both AHI and AERONET AOD exhibit a slight
increase in AOD from morning to midday. Then, AHI decreases towards the end of the
day, while AERONET stays flat. The scatter in each hourly bin is large, as shown by
error bars that span 0.6 in AOD, and thus diurnal patterns with amplitudes of 0.1,
exhibited by both AHI and AERONET fall well below a significant signal to noise
threshold. Still the basic agreement of AHI to AERONET in the overall ensemble diurnal
statistics and in the individual time series comparisons is encouraging.

**4.2 Full disk AHI-derived AOD diurnal cycle**

Previously, Figure 6 showed the mean full disk AHI AOD calculated for the approximate
times of the MODIS overpasses. Now we calculate the overall mean AHI-derived AOD
calculated over the entire day light diurnal cycle and not just at MODIS overpass time,
for the duration of our study period at each of the $0.25^{\circ}$ latitude by $0.25^{\circ}$ longitude grid
squares. Figure 9 shows this overall period mean map, with all diurnal information lost.
The period mean map at MODIS overpass time (Figure 6) looks qualitatively very similar
to the overall period mean map (Figure 9), suggesting that MODIS sampling provides a
good representation of the overall AOD distribution.


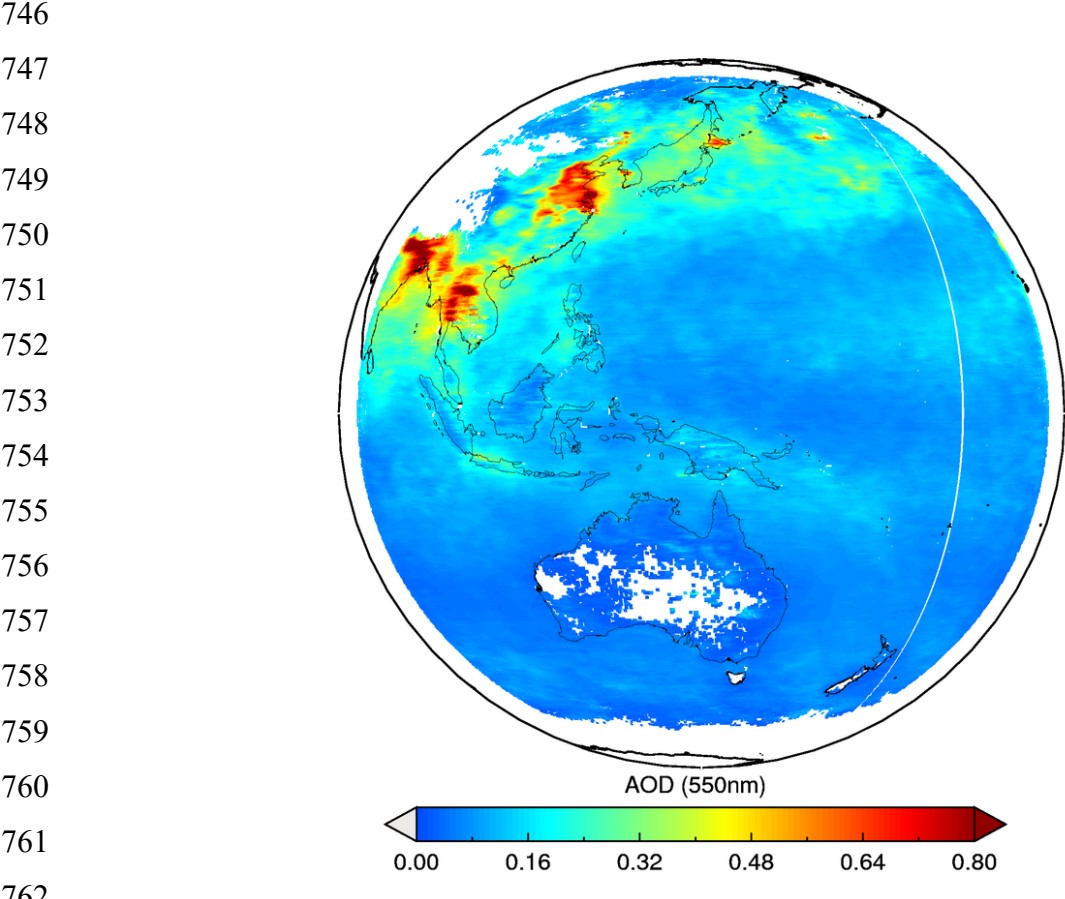


















Figure 9 Daily mean AOD at 550 nm calculated over all daylight full disk images of AHI
during May-June 2016. No requirements of collocation with MODIS or AERONET were
imposed.

Then we calculate the mean AOD for each AHI full disk scan corresponding to a
particular UTC hour, in each grid over the period of our study. Figure 10 shows the plots
of the absolute difference (mean hourly AOD minus mean daily AOD) at each of these
diurnal hours.

Figure 10 captures the diurnal signature of the aerosol over a broad region of Earth. Red
colors indicate that at a particular hour of the day, the AOD is higher than the daily mean.
Blue colors indicate that the hourly AOD is lower than the daily mean. The large gray
circle that traverses the image from hour to hour is the glint mask preventing the over
ocean algorithm from retrieving an AOD value. The glint mask is set for glint angles <
40°, which unfortunately eliminates large portions of a geosynchronous image from being
suitable for a DT aerosol retrieval. The glint mask proceeds across the image hour by
hour so that the glint mask becomes indiscernible in the daily mean. That is why there is
no apparent glint mask in the overall daily average of Figure 9, nor in Figure 6
constructed from AHI AOD collocated with MODIS. Continents and islands within the
glint mask will call on the over land DT algorithm that does not mask for glint, and
therefore, will return an AOD value.

The most striking feature in Figure 10 is the blue shading at the edges of the over ocean
retrieval domains that begin the day to the west in the Indian Ocean and then switch to
the east in the Pacific in the afternoon. This band of "lower than daily average" AOD is
associated with solar zenith angle, not view angle, as it hugs the day/night terminator in
the images, even when that terminator crosses the center of the full disk image. By 0700
and 0800 UTC, the terminator artifact encompasses a broad geographical swath of ocean,
which would introduce an incorrect interpretation of local diurnal AOD signal with
amplitudes of 0.15, when daily mean values are only 0.10. Such strong diurnal swings in
AOD over the remote ocean on global scales are unrealistic.

The problem may be introduced by the radiative transfer code used to create the Look Up
Tables for the over ocean retrieval (Ahmad and Fraser, 1982) that does not fully account
for Earth's curvature. Although this code has served the DT retrieval well through the
MODIS and VIIRs eras, those polar orbiting satellites only encounter extreme solar
zenith angles at the beginning and end of their orbits near the poles, where DT aerosol
retrievals are rare due to other factors such as extreme cloudiness or snow/ice. The
inability to properly model Earth's sphericity is likely to be of greater concern for
geosynchronous satellites that encounter extreme solar zenith angles across all latitudes
and in prime retrieval areas. See Figure 11. Currently the AHI DT algorithm retrieves all
geometries with solar zenith angle < 80 degrees. Figures 10 and 11 suggest that the
terminator artifact could be mitigated by applying a more stringent threshold of 70
degrees. However, development and application of a spherical radiative transfer code is
the more satisfying long-term solution.

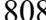


Figure 10. Difference in hourly mean AOD at 550 nm as derived from the new DT AHI
algorithm from the daily mean AOD, as plotted in Figure 8. Red indicates the specific
hour has higher AOD than the daily mean, and blue indicates the opposite.


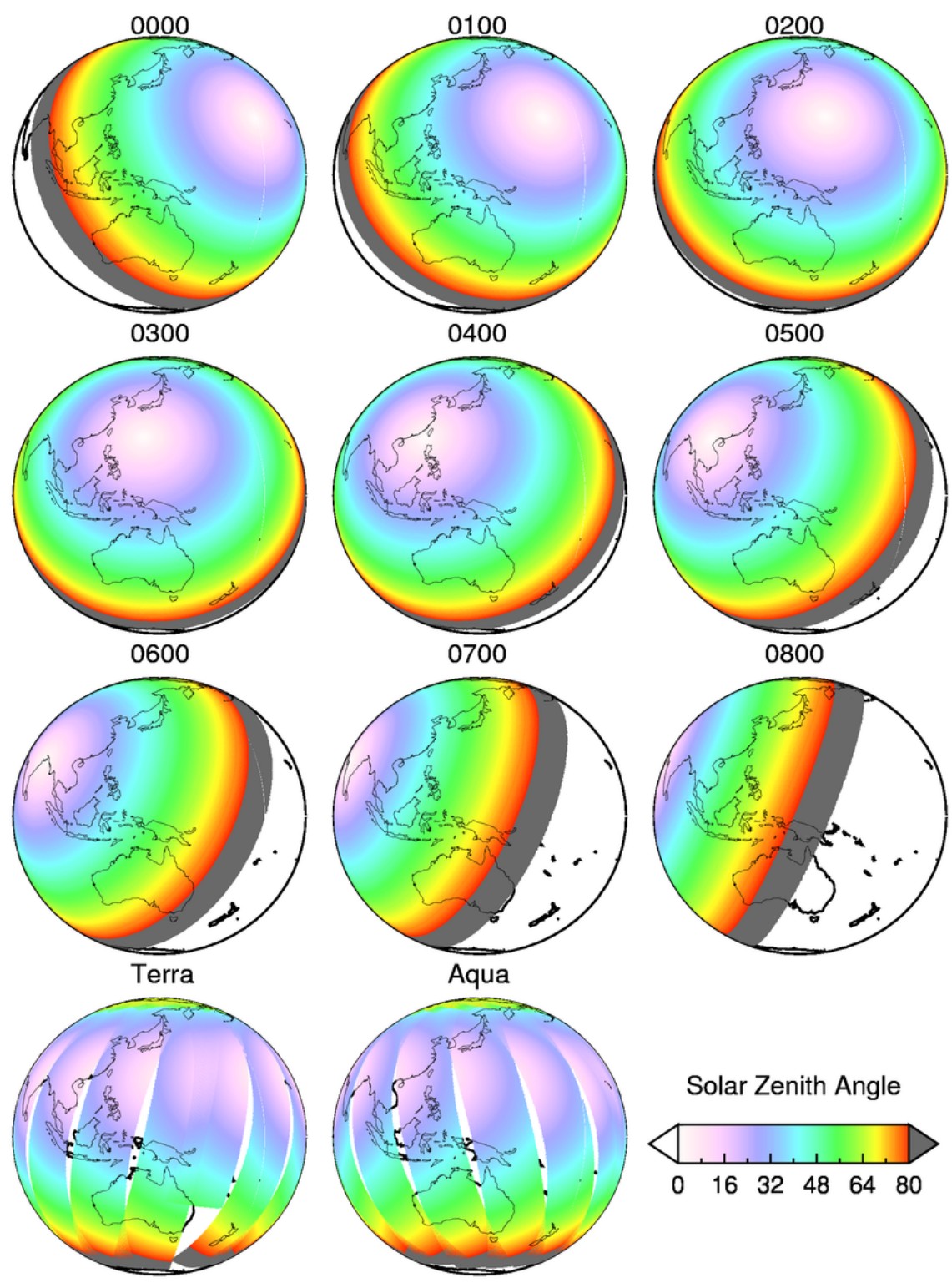



Figure 11. Mean solar zenith angle associated with each of the diurnal hours from the
AHI geometry and also for MODIS on Terra and Aqua for May 29, 2016.
There also appears to be another AOD retrieval artifact over the ocean associated with the
glint angle. Here AODs seem artificially high. Incorrect estimation of wind speed from
ancillary data or modeling of the rough ocean surface will introduce near-glint mask
inaccuracies in the AOD retrieval. With MODIS, such areas were relatively small and the
overall effect on global or regional AOD minimal. In the geosynchronous view, because
the glint mask is such a dominant feature, the near-glint artifacts appear much more
pronounced.

The good news seen in Figure 10 is that the retrieval over land does not appear to have
encountered any systematic artifacts. Blue and red shading is distributed spottedly across
the Asian, Indonesian and Australian land masses. Without validation we cannot say for
sure, but typically local factors determine aerosol diurnal trends, and thus, the spotty
blue/red shading could indicate that the retrieved AOD is representing the consequences
of these local diurnal forcing mechanisms. We have already seen in Figures 7 and 8 that
the AHI retrievals resolved the differing local diurnal patterns at three over land
AERONET stations within relative close proximity. In terms of the over land retrieval,
Figure 10 demonstrates that the DT algorithm applied to AHI will identify land regions
where the diurnal signal is more spatially cohesive. For example the east coast of India
and Bangladesh experience an increase in AOD in the late afternoon, while the overall
trend in northeast China is to decrease AOD in the afternoon, although there are local
contradictions to these regional patterns.

**4.3 AHI-derived AOD diurnal cycle over 5-degree squares**

The factors that drive a diurnal AOD signature tend to be local in character. These
include sources and sinks linked to time-of-day (rush hour traffic, agricultural burning,
afternoon convection/precipitation) or diurnally influenced mesoscale circulations and
transport (sea breeze or mountain slope regimes). Thus, individual stations as shown in
Figure 7 exhibit stronger diurnal signatures than does an ensemble average consisting of
stations distributed across the region (bottom right panel of Figure 7). The full disk plots
of Figure 10 suggest that there are regions of moderate extent that do experience a
cohesive diurnal AOD pattern. To further investigate the ability of the DT AHI to provide
insight into diurnal patterns of AOD during daylight hours we calculate the average AOD
in specific 5$^o$ latitude by 5$^o$ longitude boxes as a function of the hour of the day.

Figure 12 shows the diurnal AOD signatures of five of these 5$^o$ by 5$^o$ boxes. As suggested
by Figure 10, the AOD over northeastern China (Fig. 12, Box# 1) exhibits its highest
AOD during morning hours, 00 UTC to 03 UTC, corresponding to local times of 0800 to
1100, then experiences a slow decrease during the remainder of the day until sunset.
Average mean AOD at 550 nm in this area ranges from morning values of 0.65 to late
afternoon values of less than 0.40. Over Bangladesh (Fig. 12, Box #2) the glint mask
does not interrupt ocean retrievals until the last diurnal hour of the day. Ocean and land
retrievals exhibit very similar diurnal signatures in this area, slowly rising from morning
lows of 0.3-0.4 to late afternoon highs of 0.8-0.9, at least over land. Another area
containing both land and ocean retrievals is over northern Japan and adjacent Pacific
Ocean (Fig. 12, Box #3). This area is far enough north to not be hampered by the glint
mask at this time of year. The over ocean and over land diurnal patterns are similar with
morning to midday values of 0.30-0.35 gradually decreasing through the afternoon to
lows of 0.15 by sunset. This is a significant diurnal range of AOD over ocean.


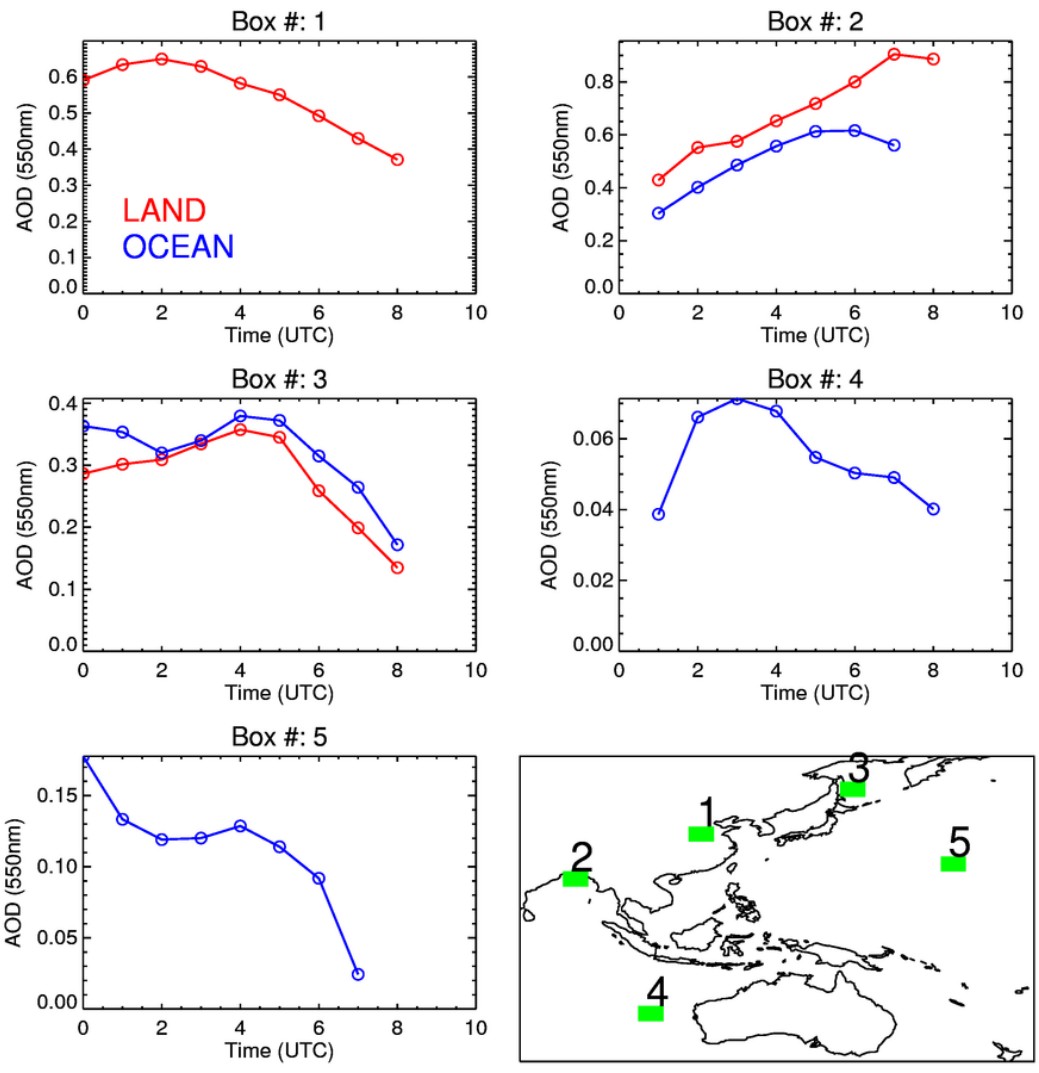

Figure 12. Spatially averaged mean AOD at 550 nm from the derived DT AHI product
for selected 5° by 5° latitude-longitude squares (boxes) in each hourly bin for the two-
month study period, producing AOD diurnal signatures for these selected areas. Red lines
depict over land retrievals. Blue lines depict over ocean retrievals. X-axes are in UTC
hours, for the reference, the local time in Beijing is +8 hours from UTC. Y-axes scale
vary from plot to plot. The green squares on global map indicate location of the specific
box.
Two areas over open ocean are shown in Figure 12, one in the Indian ocean west of
Australia (Fig. 12 Box #4) and the other in the Pacific (Fig. 12, Box #5). Note that the
scales on the y-axes of these two plots are different. At the Indian ocean area there
appears to be a diurnal signal, but the amplitude of that signal is only about 0.02, well
within the noise levels of both the retrieval itself and of the sampling and statistics of
calculating the diurnal pattern. Essentially there is no significant diurnal signal at this
location and the mean AOD is about 0.05±0.01. In contrast the Pacific example exhibits a
strong diurnal pattern, ranging almost an order of magnitude from 0.18 (~0.2) to 0.02. It
is in this area that the two ocean artifacts become apparent. During the early morning
hours this area resides just north of the sun glint mask where insufficient modeling of the
rough ocean surface creates an artifact in the retrieval, introducing a high bias. During
late afternoon hours, the solar zenith angle increases to beyond 70º and the low bias
artifact from the terminator affects the retrieval. It is only midday day when this Pacific
region escapes either artifact and then we see little diurnal signature and a mean AOD of
0.11 ± 0.015. Thus, the apparently strong diurnal signal here is in reality just the
combination of two different artifacts in the retrieval.

The examples in Figure 12 illustrate the variety of aerosol diurnal patterns over Asia with
polluted regions like northeastern China and Bangladesh showing diurnal amplitudes of
0.25 – 0.50 in AOD, but with oppositely signed slopes. The need to understand and
explain these different patterns across an area as large as Asia opens new research
questions as to what is the driving processes behind these AOD patterns, how will they
affect assimilation into global and regional models, and what are the air quality and
public health implications?  While the processes creating diurnal aerosol patterns are
primarily local, the consequences of spatially cohesive patterns will have non-local
consequences, and aerosol products from geosynchronous observations, such as the AHI
DT product, are key to identifying and quantifying these spatially cohesive situations.
The patterns seen in Figure 12 may also suffer from the caveats imposed upon the
individual station analysis of Figure 7. The diurnal patterns may be artificial constructs of
observations made at different times on different days and not represent the true change
of aerosol loading over the span of day light hours. However, because of the greater
statistical sample offered by the larger spatial domain of the 5º x 5º box there is greater
confidence in the patterns of Figure 12 than those of Figure 7.
The examples in Figure 12 also illustrate that artifacts still exist in the retrieval over the
ocean, but that not all strong diurnal signatures over ocean are due to the artifacts, as
shown in Fig. 12d where the ocean pattern mimics the artifact-free land pattern. Being
aware of the possibility of artifacts and working towards mitigating those artifacts in the
future will be essential to properly making use of any new geosynchronous product.

**5.0 Discussion and conclusions**

The traditional Dark Target (DT) aerosol retrieval algorithm was adapted for the
Advanced Himawari Imager (AHI) and applied to AHI-measured spectral reflectances
produced for a limited data set in support of KORUS-AQ for the two-month period of
May-June 2016. The adaptation makes use of the spectral similarity between AHI and its
predecessor DT sensors (e.g. MODIS, VIIRS), but omits certain important pixel selection
procedures that require spectral bands unavailable from AHI. The lack of these specific
masks may permit additional cirrus and cloud contamination into the results of this two-
month preliminary demonstration, although large-scale comparisons of collocated AHI
and AERONET or AHI and MODIS retrievals do not reveal significant overall biases.
However, AHI retrievals may be benefitting from AERONET or MODIS cloud masking
in the collocations. Expanding the AHI retrieval into the winter months when snow/ice
will be encountered will then certainly show contamination from such surfaces, as the
current DT snow/ice mask requires the 1.24 μm channel that is missing from AHI. Before
wintertime retrievals are made with AHI, a new cloud/ice mask for this sensor must be
developed.

Collocations between AHI and AERONET demonstrate that AHI retrievals match
AOD_550 nm at AERONET stations as well as the MODIS DT aerosol product matches
AERONET in terms of correlation, RMSE, overall bias and percentage within expected
error. Meeting previous MODIS DT validation criteria does not guarantee meeting the
international standards set by GCOS, as those criteria are more stringent. Additionally,
because AHI can make aerosol retrievals multiple times per day, there were
approximately twice as many AHI-AERONET collocations as there were from MODIS-
AERONET. Geostationary aerosol retrievals will significantly increase the sampling of
retrieved AOD from current polar-orbiting sensors. Not only did the DT AHI product
match AERONET, statistically, in scatterplots, it also represented the diurnal signal in
AOD, as measured by AERONET at individual stations, and in the ensemble median
statistics. The three stations shown are representative of varying retrieval biases and
exhibit different diurnal signatures, even though they are in relatively close proximity.
The AHI DT algorithm was able to distinguish these diurnal differences, although sample
size was small and signal-to-noise impeded inference of the diurnal signature.

Plotting the time series of the collocated data along the same axis shows that the AHI
AOD matches the temporal variability of the AERONET AOD hour-by-hour, even when
there is a strong bias in the magnitude. These time series plots are strong evidence that
the DT retrieval algorithm applied to geosynchronous sensors such as AHI will be able to
resolve short duration events such as individual plumes when the algorithm moves to
operational status.

Collocated AHI and MODIS retrievals demonstrate excellent agreement when applying
the DT algorithm to the two different sensors. Both AHI and MODIS produce similar
representations of the 2-month mean AOD across the AHI full disk region. However,
difference maps do show regional biases. Interestingly, AHI is overall biased low against
MODIS-Terra, but biased high against MODIS-Aqua, and thus falling within the offsets
already noted between the AODs of the two MODIS sensors (Levy et al., 2018). The one
place that AHI differs in the same way from both MODIS-Terra and MODIS-Aqua is in
its positive bias of 0.10 in the high aerosol loading regions of south, southeast and
northeast Asia. The fact that these biases are only seen in high aerosol loading suggests a
problem with the traditional DT aerosol models, not the surface parameterization. We
note that the over land aerosol models have never been tested for the unique geometry
that AHI has brought to the table.

When the algorithm is applied to the full disk image and hourly mean AOD plots are
made, we notice immediately an artifact in the diurnal signature that affects only the over
ocean retrieval. This artifact occurs at the day/night terminator and is associated with
extreme solar zenith angles, not view angles. Extreme solar zenith angles are much more
prevalent in geosynchronous images than in polar-orbiting ones, and thus our previous
experience with polar-orbiting sensors did not prepare us for this artifact. The most likely
explanation for the solar zenith artifact is the inability of the original radiative transfer
code to model spherical Earth. Earth's curvature when the sun is on the horizon will
introduce uncertainties into the radiative transfer calculation and result in inaccurate
aerosol retrievals. Until modifications can be made to the radiative transfer code, the
solution to mitigating this artifact will be to limit retrievals to lower solar zenith angles
over ocean ($<70°$). This is unfortunate because already the retrieval loses a goodly section
of the equatorial ocean because of the $40°$ glint mask when solar zenith angles are small.
Because we also saw retrieval artifacts along the edge of the glint mask, it is unlikely that
the $40°$ threshold can be relaxed. For now, the DT AHI retrieval over ocean should be
limited to a small range of solar zenith angles that will avoid both the glint and the
artifact at the terminator, and this will limit the diurnal range of AHI-retrieved AOD over
ocean.

In a preliminary analysis meant to show the scientific potential of the AHI DT product
we found a balance between the local nature of diurnal signatures and the need of a
substantial statistical sample by calculating the mean diurnal patterns of AOD in $5°$
latitude by $5°$ longitude boxes. The result of this analysis revealed a variety of diurnal
patterns across Asia, as well as illustrating diurnal patterns of ocean areas affected and
not affected by glint and solar zenith angle artifacts. A more mature AHI DT product
will enable further exploration of these diurnal patterns and the consequences these
patterns hold for climate processes, assimilation systems and air quality.

To make progress towards a more mature algorithm beyond the preliminary version
analyzed here, we will need to continue the analysis and investigate the following points:

•   What is the reason for the biases between AHI and both AERONET and MODIS?
•   Are these biases linked to solar zenith angle, view angle or scattering angle?
• Are these biases linked to surface parameterization? Specifically change in
surface ratios with viewing geometry.
• Do we mitigate artifacts by employing a more realistic spherical radiative transfer
code?
• How do we mask for snow/ice without the 1.24 μm wavelength?
• Can we characterize cloud and cirrus contamination in the retrievals, and then
mitigate those effects?
• How does the retrieved AOD spectral dependence and size parameter from AHI
compare to those from MODIS?
• Can we surpass results obtained from the polar orbiting sensors by incorporating
additional specific geosynchronous capabilities into the DT retrieval?
The short two-month demonstration described and illustrated here is a preliminary
assessment of the ability to bring the well-vetted DT aerosol retrieval to a
geosynchronous satellite sensor. The results show that porting the algorithm is possible,
that it can produce AOD that matches AERONET to the same degree as the MODIS
product, and that it can distinguish local diurnal signatures at AERONET stations over
land. The view from geosynchronous sensors will provide new insight into Earth's
aerosol system, especially if that view is steeped in and compatible with the 20-year
record of the DT polar-orbiting experience. This study puts us on the road to achieving
this new perspective.

**6. Acknowledgement**
This work was supported by the NASA ROSES program NNH17ZDA001N: Making
Earth System Data Records for Use in Research Environments and NASA's EOS
program managed by Hal Maring. We thank Space Science and Engineering Center
(SSEC), University of Wisconsin-Madison for providing Himawari-8 data. We thank
MCST for their efforts to maintain and improve the radiometric quality of MODIS data,
and LAADS/MODAPS for the continued processing of the MODIS products. The
AERONET team (GSFC and site PIs) are thanked for the creation and continued
stewardship of the sun photometer data record; which is available from
http://aeronet.gsfc.nasa.gov.

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
