# Peer review of "Applying the Dark Target aerosol algorithm with Advanced Himawari Imager observations during the KORUS-AQ field campaign 3 Pawan Gupta1, 2, Robert C. Levy3, Shana Mattoo3, 4, Lorraine A. Remer5, Robert E. Holz6, 4 5 Andrew K. Heidinger7"

_Atmospheric Measurement Techniques, 2019_

## Referee Comment (RC1) · Anonymous Referee #1 · 9 Apr 2019

This paper presents the application of the existing DT algorithm for aerosol retrieval to the Advanced Himawari Imager. The main advantage to retrieve aerosol from such an instrument is the possibility to observe the daily cycle of the aerosol load. The results presented in the paper are promising, although more effort should be spent to overcome the issues discussed in the manuscript about the missing bands and a larger data sample should be included in the validation. The paper is clear and well presented.

Here are some general comments:

L271 You often mention cloud contamination issues, saying that they are expected

in the results, but never show an example of it. It might be worth it to discuss this a bit more in depth, to quantify the impact of cloud contamination. Maybe a simple timeseries showing both your retrieval and AERONET (without any correlation) could do the job.

L283 Please specify what kind of statistical filtering is performed on the data.

L291 Please explain how the AOD at 0.55um is derived.

Figure 2: You show here 3 different situations: DT biased high, biased low and unbiased against AERONET. Could you give an interpretation of these results? What can cause this different behavior? Also, a general overestimation for low AOT is visible in Panel A of Figure 2. You should discuss where this overestimation comes from. One explanation could be the different spatial scale and the impact of residual cloud contamination at the different scale (Henderson and Chylek, 2005, Chand et al., 2012). More technical details about Figure 2 (and 4): why not to use the percentage of points satisfying the GCOS requirements instead of the EE%? The readers should be more familiar with the GCOS requirement and this will also allow an easier comparison of your performance with the ones of similar algorithms. Finally, for consistency, could you please show the regression line in Panel A of both Figure 2 and 4?

Figure 3: The figure shows that the distribution of validation statistics varies from station to station. Could you please discuss possible reasons why it happens? Do you think it is due to the land cover type and the surface reflectance parametrization? Or the aerosol type?

Some minor corrections:

L168 In this study use the full disk data is used

L271 Because alternative methods have not been developed for masking clouds, and the alternative method for identifying sediments has not been vetted to the same extent as the original MODIS DT masking techniques. Therefore, the possibility of contamination from these features affecting the aerosol retrievals is higher than expectations based on the MODIS heritage.

---

## Referee Comment (RC2) · Anonymous Referee #2 · 19 Jun 2019

In this paper, the authors ran MODIS Dark Target aerosol algorithm on Himawari-8 Advanced Himawari Imager (AHI) for two months and analyzed the retrievals for accuracy. Analysis of bias based on comparisons to AERONET and MODIS and characterizing the retrievals as a function of time of the day is presented. The AHI reflectances were averaged into a 20 x 20 square km grid to do aerosol optical depth retrieval. Matchups were done by further aggregating AHI retrievals to 0.25 x 0.25 degree grid space. The authors conclude that the performance of AHI AOD retrieval is similar to that of MODIS with some large significant biases that are not fully understood. The authors also conclude, by raising several questions in the end, that this work is preliminary and much more work needs to be done to fully transform the MODIS DT AOD algorithm to AHI
and resolve issues that arise from the differences in sun-satellite geometries.

Given the preliminary nature of this work, I encourage the reviewers to complete the work and re-submit the paper. Some chief concerns I have are: (1) A lot of spatial and temporal averaging is done rendering the advantages of a geostationary satellite useless. When the goal of a satellite in a geostationary satellite is to make frequent observations at high spatial resolution, why not demonstrate the usability of products at those resolutions? (2) Some large biases in AHI AOD vs. AERONET at some stations not explained (3) The AHI AOD artifacts near the terminator as well as sunglint region are noted. However, analysis of the bias between AHI and AERONET AOD as a function of various parameters (surface reflectance, view angle, solar angle, etc.) not carried out. The science in this paper is not new to rush towards a publication. The authors can take time to process additional data to cover different seasons and atmospheric conditions and conduct a more thorough investigations (4) An important concern that I have is that the authors do not bring in the discussion of spectral surface reflectance ratios and how the ones adapted from MODIS are suitable or not for a geostationary satellite. The viewing conditions (geometries) are quite different for geostationary satellite compared to a polar-orbiting satellite and the surface characterization need to be understood from this perspective (5) Processing data from other time periods will also provide insights into cirrus cloud contamination, pixel screening for snow/ice etc.

I applaud the work done by the authors in adapting MODIS AOD algorithm to AHI. There is no depth in the work, however. There are many groups doing similar work with much advanced state of understanding and maturity. For example, similar work done more thoroughly by other groups such as the GOCI aerosol retrievals is not mentioned in the paper (Evaluation of VIIRS, GOCI, and MODIS Collection 6 AOD retrievals against ground sunpohotometer observations over East Asia by Q. Xiao, H. Zhang, M. Choi, S. Li, S. Kondragunta, J. Kim, B. Holben, R.C. Levy, and Y. Liu, Atmospheric Chemistry and Physics, 2017). I find the work to be incomplete and only in a very

preliminary state. I recommend the authors to complete the processing to cover all four seasons and re-analyze the data to fully understand the retrievals at their native resolution and various sources of uncertainties, chief among them contributions from an inadequate characterization of surface. My recommendation, therefore, is for the paper to be resubmitted after additional work is completed.

---

## Author Comment (AC1) · 15 Sep 2019

Title changed to: "Applying the Dark Target aerosol algorithm with Advanced Himawari Imager observations during the KORUS-AQ field campaign"

**Anonymous Referee #1**

This paper presents the application of the existing DT algorithm for aerosol retrieval to the Advanced Himawari Imager. The main advantage to retrieve aerosol from such an instrument is the possibility to observe the daily cycle of the aerosol load. The results presented in the paper are promising, although more effort should be spent to overcome the issues discussed in the manuscript about the missing bands and a larger data sample should be included in the validation. The paper is clear and well presented.

Thanks for your time to review the paper and for your useful comments

**Here are some general comments:**

L271 You often mention cloud contamination issues, saying that they are expected in the results, but never show an example of it. It might be worth it to discuss this a bit more in depth, to quantify the impact of cloud contamination. Maybe a simple timeseries showing both your retrieval and AERONET (without any correlation) could do the job.

This is a good suggestion. In the new Figure 8, we plot the time series of AHI and AERONET AOD at those three stations that we introduce earlier. Interestingly cloud contamination is not obvious at these three stations in the time series. The AHI time series track the AERONET times series well, and deviations/biases could be attributed to any number of factors.

We also provide additional time series over the other 46 stations as supplement material.

Actually we don't know whether the offsets are due to clouds or not, we just want to emphasize in this paper that cloud masking was ported directly from our experience with MODIS and VIIRS, without further vetting for specific AHI applications. There is a potential for cloud effects here, but we have not proven it. To prove cloud effects and find a mitigating strategy would require a focused serious effort that is beyond the purpose of this introductory paper.

L283 Please specify what kind of statistical filtering is performed on the data.

After masking for clouds, glint, sediments, improper surfaces etc., the remaining pixels that have escaped the masking are sorted from high to low reflectance, and the darkest and brightest "good" pixels are arbitrarily eliminated. Darkest is defined as the darkest 20% over land and 25% over ocean. Brightest is defined as the brightest 50% over land and 25% over ocean.

This information has been added to the text.

L291 Please explain how the AOD at 0.55um is derived.

The over land algorithm makes use of measured reflectance at 0.47, 0.66 and 2.1 μm and assumptions about the surface reflectance to determine the aerosol loading and establish the relative weights between two aerosol models, both defined by geographical location and season. Over ocean, the algorithm uses six wavelengths (0.55, 0.66, 0.86, 1.24, 1.61 and 2.13 μm) to determine the aerosol loading and define an aerosol model from one fine mode and one coarse mode, and the relative weight between these modes. There are no restrictions on the distribution of modes by location and season in the ocean algorithm. Once the aerosol model is defined by the weighting between models or modes, the spectral extinction of the aerosol is defined. The retrieved aerosol loading can be translated to AOD at any wavelength because of the known spectral extinction, and all wavelengths are reported in the output. Spectral AOD can be described by an Angstrom Exponent, if spectral dependence is linear in log-log space. In the algorithm linearity is not required for the interpolation/extrapolation of the AOD to any wavelength. The Dark Target algorithm has been described in multiple publications, starting from Remer et al. (2005) and Levy et al. (2007), but it is described best in the on-line Algorithm Theoretical Basis Document (ATBD). Both references, and several others are given in the text.

We have also added a bit additional description of the algorithm to the text for clarity.

Figure 2: You show here 3 different situations: DT biased high, biased low and unbiased against AERONET. Could you give an interpretation of these results? What can cause this different behavior? Also, a general overestimation for low AOT is visible in Panel A of Figure 2. You should discuss where this overestimation comes from. One explanation could be the different spatial scale and the impact of residual cloud contamination at the different scale (Henderson and Chylek, 2005, Chand et al., 2012).
More technical details about Figure 2 (and 4): why not to use the percentage of points satisfying the GCOS requirements instead of the EE%? The readers should be more familiar with the GCOS requirement and this will also allow an easier comparison of your performance with the ones of similar algorithms. Finally, for consistency, could you please show the regression line in Panel A of both Figure 2 and 4?

The EE% is the standard way the DT team uses to report % of pixels falling within the uncertainty expected of the product itself.  It gives us a sense of whether a new product is meeting the standards of the original product, which in itself has become a standard within the aerosol remote sensing community.  Other groups besides the DT team has adopted this metric: NOAA (Huang et al., 2016, https://agupubs.onlinelibrary.wiley.com/doi/epdf/10.1002/2016JD024834, Sayer et al., 2014).

The GCOS requirement for AOD is 0.03 or 10%, which is more stringent than what we have been able to achieve with the DT algorithm applied to MODIS for 20 years, or to VIIRS, or now for AHI. Evaluating against the GCOS requirement is a good suggestion, and we have introduced text putting our EE% in context with the GCOS requirements.

Regression line is added in Figure 2A and 4 A.

Figure 3: The figure shows that the distribution of validation statistics varies from station to station. Could you please discuss possible reasons why it happens? Do you think it is due to the land cover type and the surface reflectance parametrization? Or the aerosol type?

We have added following text to discuss the variability in statistics.

The variability in AHI AOD performance against AERONET over the domains comes from various reasons, including variations in surface reflectance characterization (i.e. different type of land use type), variability in assumed aerosols models within the algorithm and availability of high quality valid AOD retrievals over individual stations. Often AOD is biased high when surface reflectance ratios do not conform to assumptions.  Such was the case for many years with urban surfaces, until Collection 6.1 made an alteration (Gupta et al. 2016).  Even with that alteration, DT retrievals over Beijing continue to be high (Figure 4). Low biases will occur when the assumed aerosol model is underrepresenting the amount of light absorption of the particles. The land aerosol model used in this region in this season is the moderately absorbing aerosol in May and the non-absorbing model in June. If the aerosols are actually absorbing in June or more heavily absorbing in May in a particular locality, such as at KORUS_UNIST_Ulsan, then  the retrieved AOD will be biased low. The DT algorithm is designed for global-scale representation of the aerosol system, which for GEO means full disk retrievals.  The goal is to provide the most accurate retrieval at each individual location, but the reality is that on the global scale we cannot fine-tune land surface and aerosol model assumptions for each individual location, and some locations will have products that are biased high and some biased low.

**Some minor corrections:**

L168 In this study use the full disk data is used

Revised to fix the sentence structure

[revised manuscript text omitted]

---

## Author Comment (AC2) · 15 Sep 2019

Title changed to: "Applying the Dark Target aerosol algorithm with Advanced Himawari Imager observations during the KORUS-AQ field campaign"

**Anonymous Referee #2**

In this paper, the authors ran MODIS Dark Target aerosol algorithm on Himawari-8 Advanced Himawari Imager (AHI) for two months and analyzed the retrievals for accuracy. Analysis of bias based on comparisons to AERONET and MODIS and characterizing the retrievals as a function of time of the day is presented. The AHI reflectances were averaged into a 20 x 20 square km grid to do aerosol optical depth retrieval. Matchups were done by further aggregating AHI retrievals to 0.25 x 0.25 degree grid space. The authors conclude that the performance of AHI AOD retrieval is similar to that of MODIS with some large significant biases that are not fully understood. The authors also conclude, by raising several questions in the end, that this work is preliminary and much more work needs to be done to fully transform the MODIS DT AOD algorithm to AHI and resolve issues that arise from the differences in sun-satellite geometries.

Thank you for your comments. Here just one small corrections here, for AHI & AERONET comparison, actual AHI pixel level retrievals are used to match with AERONET whereas 0.25x0.25 degree grids were used to compare with MODIS data. We have revised the text make this point clear for the reader.

Given the preliminary nature of this work, I encourage the reviewers to complete the work and re-submit the paper.

Developing an algorithm for a new sensor is a multi-stage effort, and even porting a known algorithm to a new instrument requires multiple steps. The first step in the process is to modify algorithms to receive new inputs and handle new geometry. Does the ported algorithm show any skill at all? If so, that provides motivation to continue the process. Disseminating the results of this first step is interesting to the remote sensing community who may be struggling with their own attempts to port an algorithm or develop a new algorithm for a GEO sensor. This is why we are submitting this paper now, before a final operational algorithm is achieved and before we have answers to all the questions.

We followed a similar route when describing the porting of the DT algorithm from MODIS to VIIRS. There was an initial paper published in 2015 (Levy et al., 2015) in AMT that showed significant bias with the MODIS record, and left several questions unanswered. We are following up now with a manuscript that will be submitted before the end of the calendar year that describes the final DT VIIRS operational algorithm and addresses those unanswered questions.

Remote sensing groups need to share preliminary results at all stages of development in order to advance the science of remote sensing faster. We are always very grateful to read papers by other groups that acknowledge lingering issues and show retrieval development at a preliminary stage. We feel that the work we present in this manuscript is a solid evaluation of the first attempt of porting DT to AHI. For this evaluation we leveraged the data made available during the KORUS-AQ field campaign. This was a special circumstance that allowed us to acquire AHI data and cloud mask in non-operational terms, at non-operational resolutions. This work primarily uses the data from the field campaign period and therefore the scope of the work is limited to the available data sets of that period. Furthermore, the KORUS-AQ data sets limit us to how much further we could go. We are currently working on implementing the algorithm with high resolution operationally-appropriate data, modifying cloud masks, and working to understand and answer the questions raised in this work. It may take years to move to operational status and to answer the lingering questions. In the meantime we don't want to sit on the experience we already have.

Some chief concerns I have are:

(1) A lot of spatial and temporal averaging is done rendering the advantages of a geostationary satellite useless. When the goal of a satellite in a geostationary satellite is to make frequent observations at high spatial resolution, why not demonstrate the usability of products at those resolutions?

We agree that the goal of the geostationary sensors is to obtain frequent aerosol measurements, to characterize short-lived aerosol events such as plumes, but also to characterize diurnal patterns in the aerosol that may be introduced from diurnal human, cloud or meteorological activity. Note that the DT algorithm inherently retrieves aerosol properties with a grid box that is substantially larger than the native resolution of the sensor. For MODIS with 500 m pixels, the standard retrieval grid box is nominally 10x10 $km^2$. This is not averaging. This is inherent in the retrieval to eliminate pixels that are not optimal for retrieval and still represent the spatial distribution of aerosol properties. The grouping into retrieval boxes and elimination of sub-standard pixels helps to produce a robust and accurate product. For the AHI KORUS-AQ data set that we are using, the native resolution is 2 km. Rather than following the MODIS ratio between native pixel and retrieval box, which would have produced 40x40 $km^2$ products for KORUS-AQ AHI, we have cut the size to ¼ of MODIS and implemented a 20x20 $km^2$ retrieval box. Yes it is large, but should still resolve the major aerosol events. As we move towards an operational algorithm, we will explore finer spatial resolution retrieval boxes, as we did with MODIS that now also has a 3x3 $km^2$ product. We note, that the MODIS 3 km product has proven to be less accurate than the 10 km product, and with a high bias.

The averaging to find the mean diurnal cycle is meant to discern systematic diurnal signals in the aerosol products, perhaps caused by sources or meteorology with specific diurnal signatures with the short 2-month observational period. Some examples are shown in Figure 12 and are discussed in the text. Discerning these mean diurnal signals are also a valuable contribution from a GEO sensor.

We thank the reviewer for pointing out that by seeking mean diurnal cycles, we had missed demonstrating the power of GEO to replicate diurnal variation at fine temporal scales. In the revised version we now also compare the retrieved AOD with AERONET across several days of measurements, preserving the native temporal resolution with no averaging. This plot is Figure 8.

We also provide additional timeseries over other 46 stations as supplement material (same as Figure 8). We have revised the text and title to make it clearer.

(2) Some large biases in AHI AOD vs. AERONET at some stations not explained

We purposely showed stations where the retrieval was biased, as well as one station where the retrieval was not biased. We do not want to hide the issues from the reader, but not all issues have been resolved, and this is one of the important issues that we raise. In the text we offer several suggestions as to why the bias, but we cannot make a conclusion using this limited data set.

(3) The AHI AOD artifacts near the terminator as well as sunglint region are noted. However, analysis of the bias between AHI and AERONET AOD as a function of various parameters (surface reflectance, view angle, solar angle, etc.) not carried out. The science in this paper is not new to rush towards a publication. The authors can take time to process additional data to cover different seasons and atmospheric conditions and conduct a more thorough investigations

We agree that additional analysis is required to explain certain artifacts in AHI retrieved AODs and as reviewers rightly pointed out that this required to include data from other seasons and for longer time period. Since, this paper mainly focus on our initial attempt to support a field campaign therefore, those additional analysis could not be performed. As the project move into more operational setting where routine AHI retrieval will become available, we plan to address those issues in our sub-sequent work.

We make a note about this effort in the revised text.

(4) An important concern that I have is that the authors do not bring in the discussion of spectral surface reflectance ratios and how the ones adapted from MODIS are suitable or not for a geostationary satellite. The viewing conditions (geometries) are quite different for geostationary satellite compared to a polar-orbiting satellite and the surface characterization need to be understood from this perspective

We agree with the reviewer that the surface reflectance characterization is an important aspect of the DT AOD retrieval algorithm. As discussed on (L302-305) in the manuscript, we have not modified the surface reflectance and used same as used in MODIS. We also mentioned that further research is required to accommodate change in wavelength, change in viewing geometry, and more specific to the region.

Additional text is included to identify these issues.

(5) Processing data from other time periods will also provide insights into cirrus cloud contamination, pixel screening for snow/ice etc. I applaud the work done by the authors in adapting MODIS AOD algorithm to AHI. There is no depth in the work, however. There are many groups doing similar work with much advanced state of understanding and maturity. For example, similar work done more thoroughly by other groups such as the GOCI aerosol retrievals is not mentioned in the paper (Evaluation of VIIRS, GOCI, and MODIS Collection 6 AOD retrievals against ground sunpohotometer observations over East Asia by Q. Xiao, H. Zhang, M. Choi, S. Li, S. Kondragunta, J. Kim, B. Holben, R.C. Levy, and Y. Liu, Atmospheric Chemistry and Physics, 2017). I find the work to be incomplete and only in a very C2 AMTD Interactive comment Printer-friendly version Discussion paper preliminary state. I recommend the authors to complete the processing to cover all four seasons and re-analyze the data to fully understand the retrievals at their native resolution and various sources of uncertainties, chief among them contributions from an inadequate characterization of surface. My recommendation, therefore, is for the paper to be resubmitted after additional work is completed.

As explained above, there is value in publishing at each stage in an algorithm development, even when not all questions can be answered. The data set used here cannot be expanded to a full year. It is limited to KORUS-AQ. In the future we will work with operational data sets, but analysis will take years to complete. The GOCI work and that reference in particular (Xiao et al. 2017) are very important, but none of the products in that reference were newly applied to their respective sensors. Each had a list of previous publications describing products, development and at least preliminary validation. We cannot compare the work we present here with the Xiao et al. (2017) paper. A better comparison of the stage of our work here to GOCI work would be Park et al. (2014), where the new GOCI retrieval is compared to just 2 months of collocated AERONET in conjunction with evaluating a pollution transport model. Another GOCI paper does another evaluation with AERONET using just 3 months of collocations (Choi et al., 2015). The Choi et al. (2015) goes into greater analysis, but in a very limited geographical region.  Our interest is the full disk observed by AHI. The point we want to make is that it is not unusual to publish preliminary results from a limited data set with a new algorithm, as the GOCI group demonstrates.

There is substantial analysis in the submitted study.
1.  There is preliminary validation against AERONET
2.  There is comparison with MODIS, both Terra and Aqua showing differences between AHI and MODIS is roughly the same as the differences between MODIS and MODIS
3.  (1) and (2) demonstrate that the ported DT algorithm is making overall sense when applied to AHI.  Limitations and discrepancies are noted.
4.  The full disk is evaluated, not just a small region highlighting Korea-east China-Japan
5.  The ability of the retrieved AHI product to capture the mean diurnal cycle over the 2-month observation period at specific stations is shown, so validation is not just about a scatter plot, but in the ability of the AHI product to resolve the mean diurnal cycle at a location.
6.  Now, in the new figure, the full KORUS-AQ time series shows how well the AHI product follows the AERONET time series at specific stations, with no averaging.  There are offsets, but the hour-by-hour, day-by-day signature is captured.

7. The artifact at the terminator (low solar zenith angle) is identified, and the differences in geometry between GEO and LEO (AHI and MODIS) shows that these low zenith angles were never encountered by MODIS.  Thus, moving DT to GEO is pushing DT to completely untested geometries, and those geometries have unresolved issues.  This is a significant finding.
8. The mean diurnal cycle in AOD is investigated at specific locations, some of which are obviously artifacts, but some are physical.  This demonstrates the strength of GEO to characterize physical diurnal cycles that can be linked to sources or meteorology at the regional level.
9. Finally, all of this is packaged up and discussed in a manner that provides insight to other remote sensing groups that may be trying to port their LEO algorithms to GEO, or to users who may be starting to use GEO AOD for the first time and should become aware of potential issues in the full disk retrieval.

This work is sound, substantial, innovative and important.  It should be published for the benefit of the developer and user communities.

[revised manuscript text omitted]

---

## Author Response (AR3)

**Response to Reviewers**

Referee 1: "L279 If you don't find any proof of cloud contamination, you should not attribute not good results to this cause. So, either you remove this suggestion from the text (you spend almost 20 lines on this concept), or you find a way of proving it.

The 20 lines that the reviewer identifies, beginning with L279 are part of the section where we are describing the differences in instrument configuration and how those differences may affect the application of the Dark Target retrieval on AHI. The section identifies the need for new gas absorption correction, new LUTs, and the loss of the 1.24 μm channel and potential consequences for sediment and snow masking because of that loss. It also discusses the loss of the 1.38 μm channel and potential consequences for that loss. The conclusion to these 20 lines is:

*"Because alternative methods have not been developed for masking clouds, and the alternative method for identifying sediments has not been vetted to the same extent as the original MODIS DT masking techniques, the possibility of contamination from these features affecting the aerosol retrievals is higher than expectations based on the MODIS heritage."*

We, respectfully, believe this discussion is essential information for a reader to understand all of the subsequent analysis, and that the conclusion is measured, pointing out "the possibility of contamination". In our opinion the possibility must be mentioned. We are committed to retaining these 20 lines.

Later in the conclusions (L926 to L931) we bring up the possibility again.

*"The lack of these specific masks may permit additional cirrus and cloud contamination into the results of this two-month preliminary demonstration, although large-scale comparisons of collocated AHI and AERONET or AHI and MODIS retrievals do not reveal significant overall biases. However, AHI retrievals may be benefitting from AERONET or MODIS cloud masking in the collocations."*

Here the issue is brought up and then dismissed because we did not find the biases expected from such contamination. Again, to not revisit cloud contamination, given the different sensor configuration, would be remiss in the over all presentation of the paper. We were on the look out for cloud contamination, did not find it, and therefore there can be no proof that it exists. Here we also note that contamination is difficult to identify in collocated data sets because of the assist to cloud clearing by the older sensors/algorithms.

These statements are important to the overall message of the paper, and we are committed to retaining them.

L444 GCOS requirements might be stringent, but this does not justify not using them to evaluate your product, as they set a global standard on which most of aerosol retrieval algorithm are evaluated.

Yes. GCOS does set a global standard. It is a standard that Dark Target aerosol algorithms are not yet meeting. This is stated in the revised paper in lines 449 to 454.

*"Another metric that could be used would be the Global Climate Observing System (GCOS) criteria for AOD, which is 0.03 or 10%. This is a more stringent requirement than what we have been able to achieve with the DT algorithm applied to MODIS for 20 years, or to VIIRS. Thus, the GCOS requirement is not shown on the validation plots, as it is certainty out of reach for this first test of DT applied to a GEO sensor."*

We also did a quick survey of aerosol validation papers since 2016. None of the following even mention GCOS.

https://agupubs.onlinelibrary.wiley.com/doi/epdf/10.1002/2016JD024834
https://agupubs.onlinelibrary.wiley.com/doi/epdf/10.1002/2016JD024859
https://agupubs.onlinelibrary.wiley.com/doi/epdf/10.1002/2017JD027412
https://agupubs.onlinelibrary.wiley.com/doi/epdf/10.1002/2017JD026934
https://agupubs.onlinelibrary.wiley.com/doi/epdf/10.1002/2014JD022453

The following paper does mention GCOS, but not in the sense of setting a new standard for validation and does not show GCOS criteria on any of its plots.
https://www.atmos-meas-tech.net/12/4619/2019/amt-12-4619-2019.pdf

We now cite this paper in the second revision, as it is an important new publication.

We appreciate the reviewer's interest in moving the community to an international standard for aerosol validation. However, this puts us in a bind. In this paper we are introducing a new member of the Dark Target family. In order to put this new family member into context with the 20 year Dark Target history, we have to use the same validation criteria that we have been applying since Remer et al., 2005, otherwise the community who follows DT development will have no baseline to make an evaluation. Thus, we cannot abandon the old. We could apply both the old and GCOS, but the plots get messy very quickly. We do not see the value, in this paper, of applying both criteria in the plots when we know a priori that the retrievals will not meet the GCOS criteria, have not been designed to meet the more stringent criteria and state these facts a priori on lines 449 to 454.

However, to help with the transition from old methodology to GCOS we have added a sentence at L501 to 503, and also at L941-942.

Figure3. You state that on global scale you cannot fine-tune the model and assumption at each location. Well, you are not supposed to do that. It looks here as you're justifying your results as if they don't depend on the algorithm, but they do, showing some limitations of the DT algorithm. No algorithm is perfect, it is not needed to hide limitations or current issues. Fine tuning the retrieval on specific location should never be the mean by which aerosol retrieval algorithm are evaluated."

We have tried to convey this limitation with new wording at L486 to 488 and

Referee 2: "Recommend publishing the paper with added reference to S. Kondragunta, I. Laszlo, H. Zhang, P. Ciren, A. Huff, Air Quality
Applications of Aerosol Products from the GOES-R ABI,
https://doi.org/10.1016/B978-0-12-814327-8.00017-2 in the book
The GOES-R Series: A New Generation of Geostationary Environmental Satellites 1st Edition
by Steven J. Goodman (Editor), Timothy J. Schmit (Editor), Jaime Daniels (Editor), Robert J. Redmon (Editor)"

Added

Thanks, Pawan